# On the impact of recent developments of the LMDz atmospheric general circulation model on the simulation of $CO_2$ transport

Marine Remaud[Laboratoire des Sciences du Climat et de l'Environnement],
Frédéric Chevallier[Laboratoire des Sciences du Climat et de l'Environnement],
Anne Cozic[Laboratoire des Sciences du Climat et de l'Environnement], Xin Lin[Laboratoire des Sciences du Climat et de l'Environnement], and
Philippe Bousquet[Laboratoire des Sciences du Climat et de l'Environnement]

Laboratoire des Sciences du Climat et de l'Environnement Orme des Merisiers, 91190 Saint-Aubin

**Correspondence:** Marine Remaud (mremaud@lsce.ipsl.fr)

**Abstract.** The quality of the representation of greenhouse gas (GHG) transport in atmospheric General Circulation Models (GCMs) drives the potential of inverse systems to retrieve GHG surface fluxes to a large extent. In this work, the transport of $CO_2$ is evaluated in the latest version of the LMDz GCM, developed for the Climate Model Intercomparison Project 6 (CMIP6) relative to the LMDz version developed for CMIP5. Several key changes have been implemented between the two versions; those include a more elaborate radiative scheme, new sub-grid scale parameterizations of convective and boundary layer processes, and a refined vertical resolution. We performed a set of simulations of LMDz with the different physical parameterizations, two different horizontal resolutions and different land surface schemes, in order to test the impact of those different configurations on the overall transport simulation. By modulating the intensity of vertical mixing, the physical parameterizations control the interhemispheric gradient and the amplitude of the seasonal cycle in the northern hemisphere, as emphasized by the comparison with observations at surface sites. However, the effect of the new parameterizations depends on the region considered, with a strong impact over South America (Brazil, Amazonian forest) but a smaller impact over Europe, Eastern Asia and North America. A finer horizontal resolution reduces the representation errors at observation sites near emission-hot spots or along the coastlines. In comparison, the sensitivities to the land surface model and to the increased vertical resolution are marginal.

## 1 Introduction

The accumulation of carbon dioxide ($CO_2$) in the atmosphere due to anthropogenic activity is one of the primary drivers of climate change (Ciais et al., 2013). This trace gas therefore receives particular attention and benefits from various observation networks and systems at the surface, in the atmosphere and from space (e.g. Ciais et al. (2014)). These data streams can be used to locate and quantify the sources and sinks of $CO_2$ through the inversion of atmospheric transport in a Bayesian framework. However, despite the large monitoring effort, such estimations still suffer from large uncertainties (Peylin et al., 2013). For instance, atmospheric inverse systems used in the last Global Carbon Budget of the Global Carbon Project (Le Quéré et al., 2018) disagree on the amount of the decadal land sink integrated over the northern extra-Tropical latitudes by about 1 GtC per year. Several factors could explain such an inconsistency, but uncertainties in the modelling of atmospheric transport have long

been identified as a key driver of the spread among global atmospheric inverse modelling results (Gurney et al., 2002, 2003; Basu et al., 2018).

In 1993, the Atmospheric Tracer Transport Model InterComparison (TransCom) Project was created to assess the influence of different transport algorithms on the $CO_2$ inversion problem (Law et al., 1996; Denning et al., 1999). It is still active today and has even been extended to the methane inversion problem (Patra et al., 2011). The series of TransCom and related experiments have highlighted the importance of vertical transport in this domain, with consequences on the strength of the seasonal rectifier of Denning et al. (1995), on the estimated location of the $CO_2$ sink (Stephens et al., 2007), or on interhemispheric exchange times (Patra et al., 2011). For instance, models simulating larger vertical gradients tend to show larger interhemispheric gradients in the lower troposphere (Krol et al., 2017; Saito et al., 2013). Actually, the quality of the simulated vertical transport itself is driven by various factors: horizontal and vertical resolutions, numerical diffusion, meteorological data from Numerical Weather Prediction (NWP) centres and subgrid-scale parameterizations. Numerical diffusion arises from the grid discretization and increases with coarser resolutions (Prather et al., 2008). Regarding horizontal resolution, model intercomparison experiments showed the benefit of a refined horizontal resolution to simulate the short-term variability at continental and coastal sites (Geels et al., 2007; Law et al., 2008; Patra et al., 2008; Saeki et al., 2013; Wang et al., 2016) due to finer description of orography and of the emission fluxes (Patra et al., 2008). However, uncertainties both in meteorological data and in the location of the emission hot spots limit our capacity to use higher resolution models for inversion (Lin, 2016). Even more critical are the subgrid-scale parameterizations that directly affect the simulated vertical gradient (Locatelli et al., 2015b).

The characteristics of global transport models for $CO_2$ and related tracers vary widely in the atmospheric inversion community but the models are all driven by external meteorological data, an economy of computation which is important for the simulation of the advection of a long-lived tracer like $CO_2$, and for the computation of the associated derivatives used in the inversion systems. The meteorological variables for these "off-line" models are either directly obtained from a (higher-resolution) NWP re-analysis (Olivié et al., 2004) with an appropriate interpolation procedure, or diagnosed from a NWP re-analysis (Dentener et al., 1999), or obtained from a full General Circulation Model (GCM) nudged to a NWP re-analysis (Hauglustaine et al., 2004). This list is ordered by increasing degrees of freedom on the model for the inverse modellers, but all three cases can provide a realistic representation of the synoptic patterns in the tracer fields. Here, we take the GCM of the Laboratoire de Météorologie Dynamique (LMDz, Hourdin et al. (2006a)) that, together with its off-line version, correspond to the third case, to assess the impact of the various model components on the quality of the simulation of $CO_2$, of sulfur hexafluoride ($SF_6$) and of the most stable of the radon isotopes, $^{222}Rn$. LMDz represents the atmosphere in the Earth system model of Institut Pierre-Simon-Laplace (IPSL-CM, Dufresne et al. (2013)) and as such has been contributing to the recent versions of the Climate Model Intercomparison Project (CMIP) established by the World Climate Research Program (https://cmip.llnl.gov/). Of direct relevance for our topic here, is the fact that the off-line version of LMDz (Hourdin et al., 2006b) with an associated adjoint code are used in the atmospheric inversion system of Chevallier et al. (2005), that is the current basis for the $CO_2$ inversion products of the Copernicus Atmosphere Monitoring Service of the European Commission (http://atmosphere.copernicus.eu/). As of today, the off-line model requires 44 times less CPU time than the corresponding full LMDz version.

The version of LMDz for the current CAMS inversion system was evaluated for the transport of tracers by Locatelli et al. (2015a) under the name LMDz5A. Compared to its previous off-line version, it benefited from an increased number of vertical layers from 19 to 39 and from the convective scheme of Emanuel (1991) in replacement of Tiedtke (1989). The finer vertical resolution improved the stratosphere-troposphere exchanges (STEs) that were too fast (Patra et al., 2011). The change of convective scheme increased the inter-hemispheric (IH) gradient for $SF_6$ simulations, even though this gradient remained too weak compared to observations. As a consequence, the IH gradient in methane emissions estimated through inverse modelling is smaller compared to the inversion based on the Tiedtke (1989) convective scheme (Locatelli et al., 2015b).

Since Locatelli et al. (2015a), new versions of the full LMDz GCM have been developed, for instance for the ongoing CMIP6. The latter benefits from a resolution increased to 79 vertical layers and a more elaborate subgrid scale parameterizations in terms of convection and boundary layer processes. This version has been primarily developed for climate modelling and has not been tested yet for the transport of tracers such as $CO_2$. In this context, the objectives of this paper are twofold:

1. Evaluate the effect of these new developments on the simulated values of $CO_2$ mixing ratios and, to a smaller extent, $SF_6$ and $^{222}Rn$ mixing ratios and anticipate their benefit for inverse modelling.

2. Benchmark the sensitivity of tracer transport to model setups. Different from a multi-model intercomparison experiment, this study provides an opportunity to focus on some model components separately.

In Section 2, we describe the various LMDz configurations, the observations and the analysis methods used in this study. In Section 3, we focus on the general behaviour of the simulations, considering zonal mean features and the total column of $CO_2$. In Section 4, we compare the simulations with surface and aircraft $CO_2$ measurements. Section 5 is the conclusion.

## 2    Data and methods

### 2.1    Model description

We focus here on two reference versions of LMDz that were prepared for past ($5^{th}$) and ongoing ($6^{th}$) versions of the CMIP program. In addition to different spatial resolutions, these two versions use different subgrid-scale parameterizations or physics called 5A and 6A.

In the LMDz5A version (Hourdin et al., 2006a), turbulent transport by eddies in the boundary layer is represented by a vertical diffusion scheme where the turbulent diffusion coefficient depends on the Richardson Number (Louis, 1979). A counter-gradient term on potential temperature (Deardorff, 1972) was added to handle dry convection cases in the boundary layer. Deep convection is parameterized by the episodic mixing and a buoyancy sorted scheme of Emanuel (1991) in which both triggering and closure of the updraft depend on the potential convective energy available over the column (CAPE). These assumptions are based on the Quasi-Equilibrium (QE) hypothesis that stipulates that all convective instability available in the column is consumed instantly by deep convection that, in return, brings it back to neutral stability. The known weaknesses of this physics include: the underestimation of shallow convection (Zhang et al., 2005) resulting in insufficient venting of the

boundary layer tracers by cumulus (Locatelli et al., 2015a), the unrealistic phasing of the diurnal cycle of convection over continents, the precipitation peak being generally simulated too early in the day (Guichard et al., 2004), and the lack of tropical variability (Lin et al., 2006).

In order to address these deficiencies, a new version of the LMDz GCM, called LMDz5B, has been developed for CMIP5 (Hourdin et al., 2013). The new physics treats shallow and deep convection separately. On the one hand, shallow convection is represented in a unified way by combining the diffusive approach of Mellor and Yamada (1974) for the small scale turbulence and a mass flux scheme, the thermal plume model (Rio and Hourdin, 2008), that represents both dry and cloudy thermals in the convective boundary layer. On the other hand, deep convection and downdrafts are represented by the Emanuel (1991) scheme coupled with a parameterization of cold pools (Grandpeix et al., 2009). Deep convection triggering and closure are not CAPE functions anymore. They depend on sub-cloud processes. The convective onset is now controlled by the thermal plume variables and the maintenance of deep convection after its onset is operated by the cold pools. In better agreement with observations, the main results are a delay of the convective initiation, a self-sustainment of convection through the afternoon (Rio and Hourdin, 2008; Rio et al., 2009) and a drastic increase of the tropical variability of precipitation (Hourdin et al., 2013). This version has not been implemented in the above-mentioned inversion system for $CO_2$ because preliminary $CO_2$ transport simulations showed unrealistically large seasonal cycles at some southern stations like Palmer Station (PSA) in Antarctica (unpublished results). However, it was successfully used for aerosol data assimilation around North Africa by Escribano et al. (2016) and showed promising improvements for the representation of the magnitude of diurnal variations of surface concentrations (Locatelli et al., 2015a).

For CMIP6, configuration 5B of LMDz has further evolved from Hourdin et al. (2013): it has a different formulation of the triggering assumptions, a different radiative transfer code and it accounts for the thermodynamical effect of ice. The convective triggering is now based on evolving statistic properties on the thermal plumes by considering a thermal size distribution instead of a bulk thermal (Rochetin et al., 2013). The motivation behind this change was to depart from the QE hypothesis and to allow a more gradual transition between shallow and deep convection through three step processes (appearance of clouds, crossing of the inhibition layer, and deep convection triggering). In the shortwave, the code is an extension to 6 bands of the initial 2-band code that is used in LMDz5A (Fouquart and Bonnel, 1980), as implemented in a previous version of the ECMWF numerical weather prediction model. In the longwave, LMDz uses the Rapid Radiation Transfer Model (RRTM) (Mlawer et al., 1997). This version is now called 6A.

For the energy and water flux between land surface and atmosphere, LMDz can be coupled with the ORCHIDEE (ORganizing Carbon and Hydrology In Dynamic Ecosystems, version 9) (Krinner et al., 2005) terrestrial model or to a simple bulk parameterization of the surface water budget.

The reference configuration of LMDz5A used in CMIP5 had 39 eta-pressure layers and $96 \times 96$ grid-points, i.e. a horizontal resolution of 1.89°in latitude and 3.75°in longitude. Current reference simulations of IPSL-CM for CMIP6 use the new configuration of LMDz6A with a refined grid of 144 grid points both in latitude and longitude directions and a vertical resolution extended to 79 layers. The number of layers under 1 km has increased from 5 to 16 layers. The remaining additional layers are mostly located in the stratosphere so that in the lower stratosphere (between 100 and 10 hPa), the vertical spacing $\Delta z$ is

approximately 1 km in this model setup. For the inverse system, LMDz is currently run in an offline version of configuration 5A with 39 eta-pressure layers and $96 \times 96$ grid-points, i.e. a horizontal resolution of $1.89°$ in latitude and $3.75°$ in longitude.

## 2.2 Description of the simulations

We have run the two versions of the physics described above, 5A and 6A, at several resolutions for the years from 1988 to 2014. A summary of the simulations used is given on Table 1. The identification number of the LMDz code used here (that contains both physics versions) is 2791. We discard the first 10 years (1988-1998) to allow enough spin-up for the tracer simulations, considering the interhemispheric exchange time of about 1 year for passive tracers (Law et al., 2003). The dynamics is nudged towards the 6-hourly horizontal winds from the ECMWF reanalysis (Dee et al., 2011) with a relaxation time of 3 hours (Hourdin and Issartel, 2000). $CO_2$, $SF_6$ and $^{222}Rn$ initial values are set uniformly for all model grid boxes at a value of, respectively, 350 $\mu$mol/mol (abbreviated as ppm), 1.95 pmol/mol (abbreviated as ppt) and 0 $Bq/m^3$ on 1 January 1988. 350 ppm is the global mean given for that date by the forward simulation associated to the CAMS $CO_2$ inversion used here (see Section 2.3). 1.95 ppt is the initial value used for $SF_6$ in the TransCom protocol of Denning et al. (1999). The initial value of $^{222}Rn$ does not matter given the short lifetime of this radionuclide. The time step of model output is hourly.

| Version | Physics | Horizontal resolution (longitude×latitude) | | Number of |
|---|---|---|---|---|
| | | degrees | points | vertical levels |
| 5A-96L39 | 5A (old physics) + ORCHIDEE | $3.75° \times 1.90°$ | $96 \times 95$ | 39 (260 m) |
| 6A-96L39 | 6A (new physics) + ORCHIDEE | $3.75° \times 1.90°$ | $96 \times 95$ | 39 (260 m) |
| 6AWOR-96L39 | 6A (new physics) + bulk model | $3.75° \times 1.90°$ | $96 \times 95$ | 39 (260 m) |
| 6A-144L39 | 6A (new physics) + ORCHIDEE | $2.50° \times 1.30°$ | $144 \times 143$ | 39 (260 m) |
| 6A-144L79 | 6A (new physics) + ORCHIDEE | $2.50° \times 1.30°$ | $144 \times 143$ | 79 (670 m) |

**Table 1.** Description of the simulations. The vertical spacing averaged over the tropics (between the surface and 10 km) is indicated in brackets.

Numerical approximations in the advection scheme and in the sub-grid parameterizations prevent LMDz from strictly conserving mass. For $CO_2$, for instance, the model loses about 1 GtC integrated over 10 years for the reference version and twice as much for the new version. We have therefore applied a global mass correction both on the $CO_2$ and the $SF_6$ 3-dimensional mole fraction fields every hour. The correction method consists in a diagnostic of the loss at each timestep which is then added back evenly distributed through space.

## 2.3 Prescribed tracer fluxes at the surface

$CO_2$ surface fluxes are prescribed every 3 hours from version 15r4 of the $CO_2$ atmospheric inversion product of the CAMS. The 3-hourly resolution is allowed by prior information to the inversion system, while surface air sample measurements constrained the fluxes at weekly or coarser resolution (they also correct a mean day-night difference every day but this is marginal). The inversion system assimilated the surface measurements for the period 1979-2015 in an off-line version of LMDz5A at hor-

izontal resolution 3.75°×1.90°(longitude × latitude). Of interest here is the use of fossil fuel emissions from the Emission Database for Global Atmospheric Research version 4.2 (EDGAR, http://edgar.jrc.ec.europa.eu/) scaled to the annual global values of the Global Carbon Project (Le Quéré et al., 2015). Details of the prescribed fluxes are given in Chevallier (2017). As a consequence, the surface fluxes carry some imprint of a version of LMDz close, but not identical, to 5A-96L39. Fluxes from another atmospheric inversion could have been used instead, but recall that the most robust atmospheric inversions share the same surface measurements to a large extent, so that the question of the lack of independence of our $CO_2$ simulations to the surface measurements would remain anyway. In the supplement, we show the robustness of our conclusions with respect to a change of the $CO_2$ surface fluxes from CAMS to CarbonTracker (https://www.esrl.noaa.gov/gmd/ccgg/carbontracker/).

For use at resolution 2.50°×1.30°, the natural component of the optimized fluxes has been interpolated from its native 3.75°×1.90°resolution, and has been completed by a fossil fuel component directly interpolated from the EDGAR native 0.1°×0.1°resolution in order to avoid artificial smoothing. All grid changes here conserve mass.

Monthly averages of $SF_6$ emission fluxes at 1°×1°are taken from the EDGAR 4.0 inventory for the period 1988-2008 as corrected by Levin et al. (2010). The global emissions steadily increased from 934 mmol/s in 1988 to 1599 mmol/s in 2010. Since these sources are mostly in the northern hemisphere and since there are no sinks, $SF_6$ has been largely used to gain further insight into IH transport and STEs. We additionally prescribe $^{222}Rn$ surface fluxes according to Patra et al. (2011). With its short lifetime (3.8 days), $^{222}Rn$ is used here to gain some insight into the vertical mixing within the column.

### 2.4 Observations and data sampling

### 2.4.1 Model sampling strategy

For each species, the simulated concentration fields were sampled at the 3D grid boxes nearest from observation location. They were also sampled to the nearest hours from the time when the observations were taken. Observations are all dry air mole fraction measurements calibrated relative to the $CO_2$ World Meteorological Organization (WMO) mole fraction scale. For comparison, the corresponding dry air variables in the model simulations are used. In Section 3, even though the model simulations are not compared to measurements, the model sampling still refers to some observation selection (in the afternoon for the zonal-mean profiles, or following a satellite retrieval pattern for the total column), as indicated in the corresponding text.

### 2.4.2 Point samples from surface sites

The surface measurements of $SF_6$ are taken at background stations SPO, PSA, CGO, SMO, RPB, MLO, MHD, BRW, SUM, ALT from the NOAA/ESRL network (http://www.esrl.noaa.gov/gmd).

The simulated mole fractions of $CO_2$ were compared with some of the atmospheric surface measurements that were assimilated when optimizing the surface $CO_2$ fluxes prescribed here. The location of these assimilated surface stations is shown in Figure 1. As in the CAMS inverse modelling framework, we retain only early afternoon data (12:00-15:00 LST) for continuous stations under 1000 m.a.s.l and night-time data (00:00-3:00 LST) for continuous station above 1000 m.a.s.l. All measure-

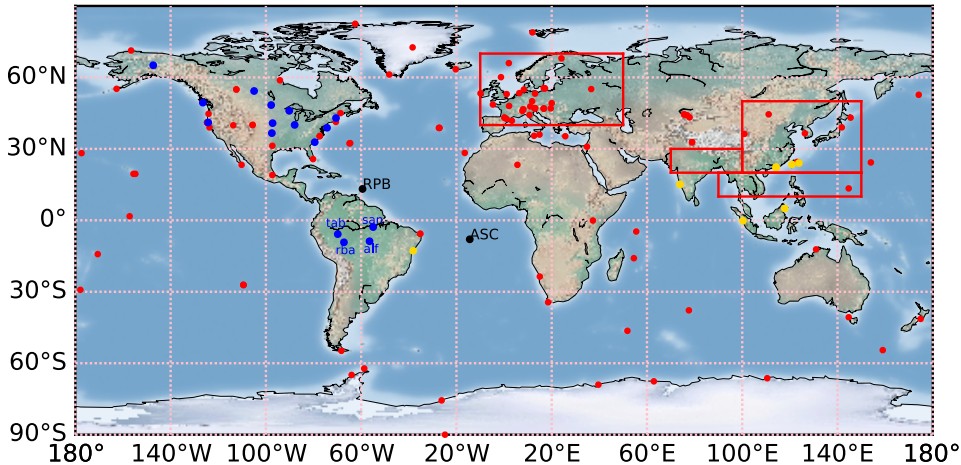

**Figure 1.** $CO_2$ sampling locations. Red dots denote the subset of the assimilated site locations that are used here. Yellow dots denote unassimilated site locations. Blue dots denote independent aircraft measurement locations in America (other aircraft sites for the rest of the world are shown in Figure 2). Specific areas for our study are shown in red: Europe (EUR: 40-70° N, 10° W-50° E), Greater Northern India (IND: 20-30° N, 70-100° E), East Asia (EAS: 20-50° N, 100-150° E), Northern Southeast Asia (NSA: 10-20° N, 90-160° E). Stations RPB and ASC, in black even though they have been assimilated, are the NOAA tropical Atlantic sites used to define the background concentrations of $CO_2$ and $SF_6$ coming into the Amazon basin.

ments from flasks below 1000 m.a.s.l have been kept. The reasons behind this hour selection are the failure of transport models in general to accurately represent the accumulation of tracers near the surface at night and the advection of air masses during the day by upslope winds over sunlit slopes in the afternoon (Geels et al., 2007). A description of the surface observations used for the inversion can be found in Chevallier (2017), but only a subset is used here. This subset comes from the obspack_co2_1_GLOBALVIEWplus_v3.2_2017-11-02 archive (Cooperative Global Atmospheric Data Integration Project, 2017), from the World Data Center for Greenhouse Gases archive (https://ds.data.jma.go.jp/gmd/wdcgg/) and from the Réseau Atmosphérique de Mesure des Composés à Effet de Serre monitoring network (https://www.lsce.ipsl.fr/). We have selected the sites with more than 3 years of record and with enough data density in time to compute the statistics.

In addition, we use some unassimilated surface observations in the Tropics (bkt, cri, hkg, hko, lln, hat - note the lower letter case used here to denote sites unassimilated in the Chevallier (2017) inversion) to better evaluate the quality of the inversion over the tropics which are not well constrained. We sampled the model output at the elevation (above sea level) corresponding to the actual elevation of each site. hkg and hko only provide the daily mean mole fraction of $CO_2$.

### 2.4.3 Vertical profile samples from aircraft measurements

We have compared the simulated $CO_2$ mole fractions against observations of $CO_2$ vertical profiles from three sampling programs: Comprehensive Observation Network for TRace gases by AIrLiner (CONTRAIL), the NOAA/ESRL Global Greenhouse Gas Reference Network Aircraft Program and the lower-tropospheric greenhouse gases sampling program over the

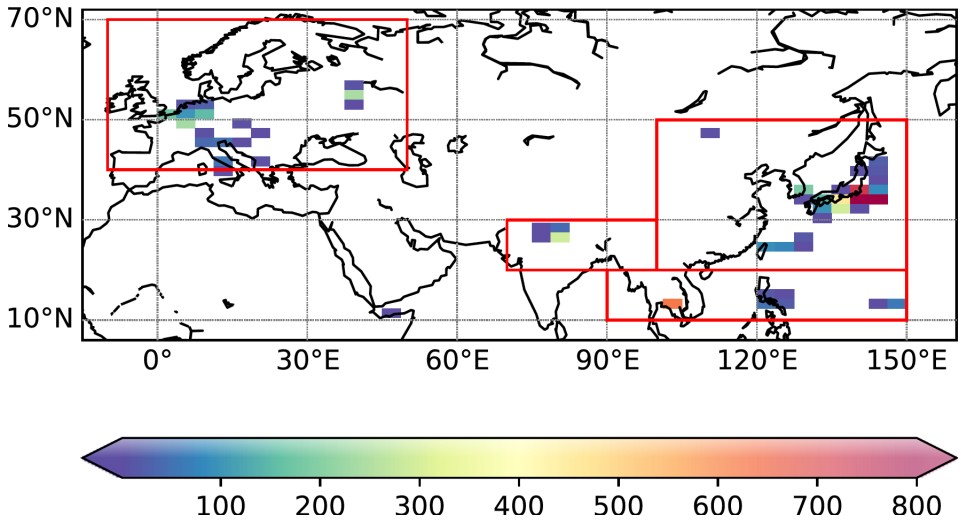

**Figure 2.** Number of CONTRAIL measurements used here at 5.5 km above sea level, within the model grid boxes ($3.75° \times 1.90°$). The specific areas of Figure 1 are also shown. Prior to the calculation of this number, the measurements have been averaged hourly in each grid box.

Amazon described in Gatti et al. (2014). Aircraft measurements have not been assimilated in the CAMS inversion product and are therefore called *independent* in the following.

CONTRAIL (Machida et al. (2008), http://www.cger.nies.go.jp/contrail/index.html) provides high-frequency $CO_2$ measurements over 43 airports worldwide and during commercial airflights between Japan and other countries. The calibration of the data is assured within 0.2 ppm (Machida et al., 2008). We selected from the CONTRAIL dataset all the $CO_2$ vertical profiles during the ascending and descending flights for the period 2006-2011 over the regions portraited in Figure 1. The regions are similar to Niwa et al. (2011) and have been chosen according to the number and location of the vertical profile samples. The number of hourly-mean measurements at 5.5 km a.s.l. per model grid box are shown in Figure 2 considering a model resolution of $3.75° \times 1.90°$. There are 862 hourly-mean vertical profiles over EUR, 4124 over EAS, 265 over NSA, 153 over IND.

The NOAA/ESRL Global Greenhouse Gas Reference Network Aircraft Program consists here of measurements of air samples collected every few days or months at 22 aircraft profiling sites over continental North America (shown in blue in Figure 1) between altitudes 300 and 8000 m.a.s.l. In the lowest altitudes, compared to the CONTRAIL measurements that have been sampled nearby commercial airports, these measurements are not affected by local emissions. We performed statistics on 974 available vertical profiles.

The lower-tropospheric greenhouse base sampling program over the Amazon provides biweekly air sample profiles from above the forest canopy (300 m) to 4.4 km above sea level at 4 sites (san, tab, alf and rba) in 2010. The locations of the airborne platforms are shown in blue in Figure 1. During their descending flights, small aircrafts filled flasks between 12:00 and 13:00 LST when the boundary layer is fully developed. Most of the samples are representative of air masses that have been blown away by the dominant easterly flow from the tropical Atlantic ocean across the Amazonian Basin. Air masses at sites tab and

rba are mainly related to transport of source and sinks from a large fraction of the Amazonian forest. Air masses at Alta Floresta (alf) and Santarem (san) are related to transport of sources and sinks of savanna and agricultural lands within their footprint areas. This aircraft data is fully described in Gatti et al. (2014) and available at ftp://ftppub.ipen.br/nature-gatti-etal/.

## 2.5   Post-processing of the $CO_2$ simulations and measurements

In Section 4, the features of interest (annual mean, monthly smooth seasonal cycle, synoptic variations) are derived from the surface data using the CCGVU curve fitting procedure developed by Thoning et al. (1989) (Carbon Cycle Group (Earth System Research Laboratory (CCG/ESRL), NOAA, USA) and following the setup of Lin et al. (2017). The CCGVU procedure is fully described and freely available at http://www.esrl.noaa.gov/gmd/ccgg/mbl/crvfit/crvfit.html. The procedure estimates a smooth function by fitting the $CO_2$ time series to a first order polynomial equation for the growth rate combined with a two-harmonic

function for the annual cycle. The seasonal cycle and annual gradient are extracted from the smooth function while the synoptic variability is defined from a residual time series between the smooth function and the raw time series. In addition, outliers are discarded if their values exceed three times the standard deviation of the residual time series.

For each station, the annual gradient to MLO is calculated by subtracting the annual mean of the $CO_2$ mole fraction at MLO (Mauna Loa, 19°52' N 155°58'W) to the annual mean from the smooth curve of the station of interest. Regarding the seasonal

cycle, the amplitude is calculated from the smooth curve as an absolute peak-to-peak difference within a year at each site. Then, we average these yearly amplitudes over the period 1998-2014. The seasonal phase is evaluated using the Pearson coefficient between observed and simulated smooth curves. The synoptic curve is extracted at each site from the residual between the raw time series and the smooth curve. In order to plot the seasonal latitudinal gradient of $CO_2$, we choose marine boundary layer sites : ZEP (Zeppelin, Ny-Alesund, Svalbard, Norway and Sweden), ICE (Storhofdi, Vestmannaeyjar, Iceland), SHM (Shemya

Island, Alaska, USA), AZR (Terceira Island, Azores, Portugal), MID (Sand Island, Midway, USA), MNM (Minamitorishima, Japan), KUM (Cape Kumukahi, Hawaii, USA), GMI (Mariana Islands, Guam), CHR (Christmas Island, Republic of Kiribati), SMO (Tutuila, American Samoa), CGO (Cape Grim, Tasmania, Australia).

The synoptic variability is evaluated using two quantities: the Pearson correlation coefficient and the model-to-observations ratio of the standard deviation (Normalized Standard Deviation, NSD) between the observed and simulated residual time

series. For each site, the diurnal amplitude is calculated from a residual time series between the raw time series of the $CO_2$ mole fraction and its daily mean.

For the airborne measurements from the NOAA/ESRL Global Greenhouse Gas Reference Network and from CONTRAIL, only the $CO_2$ samples taken in the afternoon (between 11:00 and 20:00 LST) have been retained. The resulting samples have been averaged into vertical bins of 1 km for each hour, before being averaged spatially for a given region of Figure 1 and

monthly. For each subregion and each 1-km altitude bin, a detrended signal at 3.5 km has been subtracted to the time series. Over the Amazon, a background time series has been subtracted to the simulated and observed vertical profiles through the same method described in Gatti et al. (2014).

# 3 General behaviour

## 3.1 Zonal mean structures

We first study the zonal-mean structure of the $^{222}Rn$, $SF_6$ and $CO_2$ simulations. We focus on the boreal summer (JJA) as the convection is more active over Northern Hemisphere continents during this season and the spread among the versions is the largest. Figure 3(a) shows the vertical structure of the zonal-mean mole fraction of $^{222}Rn$ from 5A. $^{222}Rn$ is a short-lived radioactive tracer naturally emitted by continental surfaces that decays radioactively with a half time of 3.8 days. Its lifetime is comparable to that of mesoscale convective systems over the tropics (10 hours on average but it can reach 2-3 days (Houze, 2003)). For this reason, $^{222}Rn$ has been largely used by modellers to evaluate vertical transport operated by subgrid-scale processes in the PBL and low troposphere (Genthon and Armengaud, 1995; Belikov et al., 2013). The vertical profile, with a maximum at ground level and a decrease with increasing height, mainly reflects the transport by convective processes between $10°$ N and $70°$ N from the boundary layer to the tropopause.

Figure 3(d) shows that the effect of the modified physics is a radon depletion with respect to 5A over the entire mid-troposphere above 7.5 km, between $30°$ S and $80°$ N. The largest relative depletion, of about half of the $^{222}Rn$ concentrations in 5A, occurs in the northern mid-latitude troposphere around 10 km. The lower concentrations of $^{222}Rn$ suggests that there is, on average, less convection penetrating into the upper troposphere in the new physics. However, the increase of $^{222}Rn$ at 2.5 km and the decrease at the surface manifests from the thermal activity that transport tracers from the surface to the top of the boundary layer. The mean reduction in active convection over the continents shown by the $^{222}Rn$ mole fraction suggests that an effect of the stochastic triggering based on thermal activities is to prevent the triggering of spurious deep convection. This observation is consistent with previous findings that thermal activities reduce the strength of the deep convection (Rio et al., 2009; Locatelli et al., 2015a). The land-surface model (Figure 3 (g)), the horizontal resolution (Figure 3 (j)) and the vertical resolution (Figure 3 (m)) have a modest effect on the vertical structure of $^{222}Rn$ compared to the physics. They enhance (land-surface) or attenuate (vertical resolution) the changes induced by the new physics in the northern mid-latitudes. For instance, Figure 3 (m) shows a slight increase around 10 km (10% of the total concentration), meaning that more deep convection penetrates within the upper troposphere with a finer vertical resolution.

$SF_6$ is a quasi-inert gas released into the atmosphere by electrical and metal industries (Maiss et al., 1996). Because of its quasi-inert nature (lifetime around 1000 years (Ravishankara et al., 1993; Morris et al., 1995; Kovács et al., 2017; Ray et al., 2017)) and its weak seasonality, we use $SF_6$ to gain insight into the large-scale transport in our simulations. Figures 3 (f), (i), (l), (o) highlight the effects of the model setups described earlier on the zonal mean distribution of $SF_6$. The modified subgrid-scale parameterization has much more impact on the zonal mean of $SF_6$ in the stratosphere than in the troposphere. The stratosphere is not as mixed as the troposphere, resulting in a longer exchange time scale and in an integration of the differences over time. The higher mole fraction of $SF_6$ means the air is younger, suggesting an accelerated Brewer Dobson circulation. The effect of the physical parameterizations on the STE fluxes has also been noticed by Hsu and Prather (2014), using two cycle versions (with two physics) of the ECMWF fields as an input to their offline transport model. The cause of this modified stratospheric dynamics is unclear and worth further investigation. Out of the stratosphere, differences between

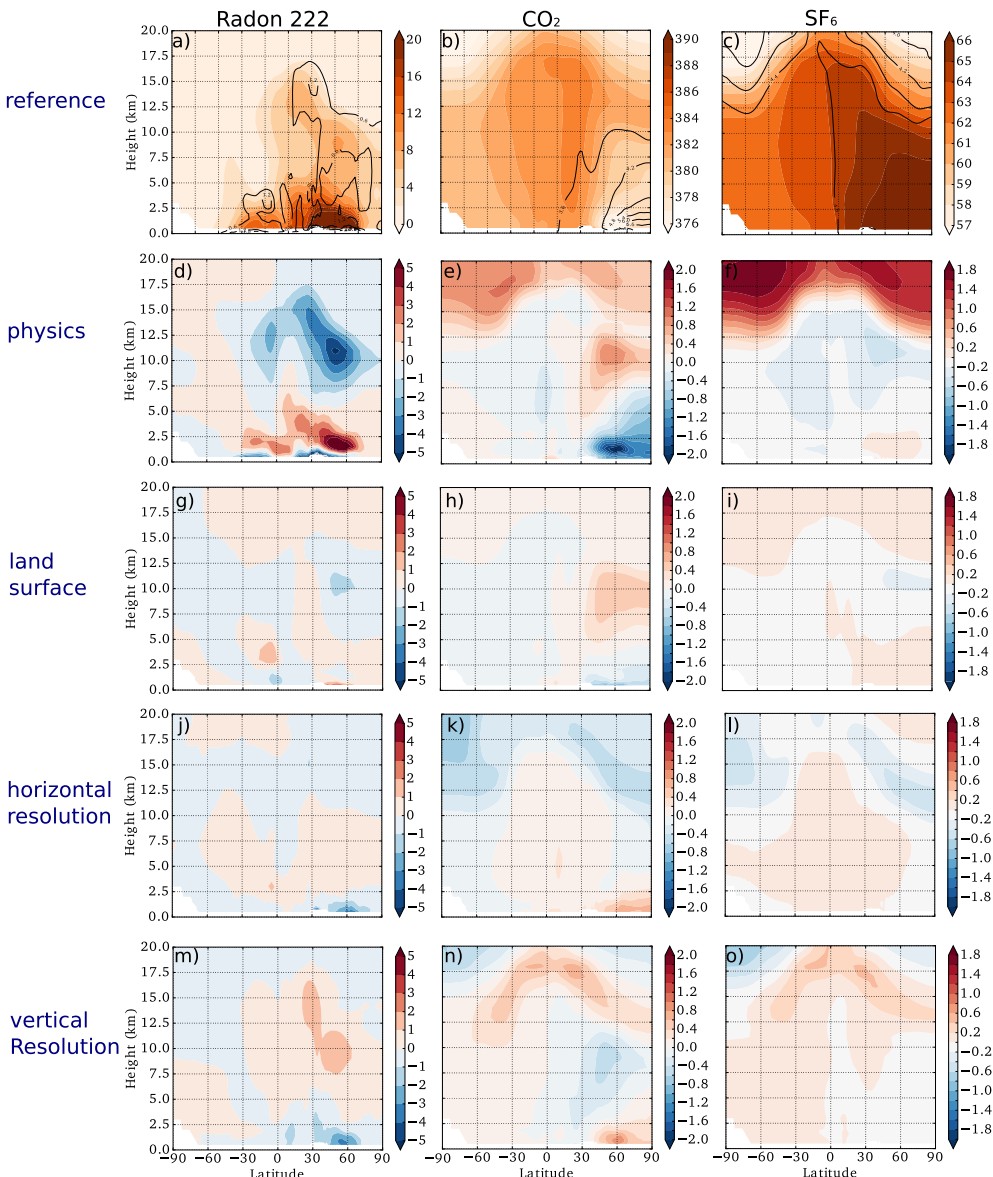

**Figure 3.** Zonal-mean mole fraction of (a) $^{222}Rn$ in $10^{21}$ mol/mol, b) $CO_2$ in ppm, (c) $SF_6$ in 0.1 ppt from 5A-96L39. The standard deviation is superimposed with contour lines. (d-f) Zonal-mean mole fraction difference between 6A-96L39 and 5A-96L39 (effect of the new physics). (g-i) As (d-f) but between 6A-96L39 and 6AWOR-96L39 (effect of the land surface model ORCHIDEE). (j-f) As (d-f) but between 6A-144L39 and 6A-96L39 (effect of the horizontal resolution). (g-i) As (d-f) but between 6A-144L79 and 6A-144L39 (effect of the vertical resolution). The zonal mean is calculated for afternoon hours from 2005 to 2010 in summer (JJA).

simulations are, on the whole, small. By comparison, they are within the compatibility range of 0.02 ppt recommended by the WMO for the surface measurements from different laboratories (Global Atmosphere Watch, 2015). The negative anomaly at

10 km is an exception. It reaches -0.06 ppt in Figure 3 (f) in the northern mid-latitudes, consistent with a less efficient vertical mixing induced by the new physics. In the northern hemisphere, the positive anomaly of 0.02 ppt in the boundary layer reveals an increase both of the surface latitudinal gradient and of IH exchanges. The strength of the latitudinal gradient of $SF_6$ is a good indicator of IH exchange as emissions are mainly located in the northern hemisphere. The increase of the latitudinal

gradient along with a weaker vertical mixing is consistent with Krol et al. (2017) who showed that the IH transport time scale is negatively correlated with the efficiency of vertical mixing and, hence, to the parameterization of subgrid scale processes.

Contrary to $SF_6$, the zonal mean distribution of $CO_2$ exhibits a strong seasonality in the northern mid-latitudes. In boreal winter, the prevalence of the fossil fuel emissions along with stable boundary layer conditions contribute to increase $CO_2$ in the boundary layer. In boreal summer, the $CO_2$ sink by photosynthesis outweighs fossil fuel emissions and terrestrial sources

(respiration, land-use), leading to a net drawdown of $CO_2$ mole fraction at the surface as seen in Figure 3 (b) beyond 50° N. As a result, the effect of the physics has an opposite sign on the $CO_2$ distribution compared to $SF_6$: a negative anomaly greater than 1.5 ppm in the PBL and a positive anomaly of 0.5 ppm around 10 km. The new physics amplifies the trapping of negative anomalies of $CO_2$ near the surface, consistent with a less efficient vertical transport. Within the lower stratosphere, the accelerated Brewer Dobson circulation induced by the new physics results in an increase of 1 ppm, which represents about

one quarter of the seasonal variability of $CO_2$ between 16 and 17 km (Diallo et al., 2017). The land-surface and resolution have a modest impact on the vertical distribution of $CO_2$.

### 3.2   Simulated x$CO_2$ convolved with the OCO-2 space-time coverage

In a similar way to the zonal-mean distribution, we analyze the seasonal climatology of the column-average dry air mole fraction of $CO_2$, denoted x$CO_2$, convolved with the space-time coverage of NASA's retrievals from the Second Orbiting

Carbon Observatory (OCO-2, Eldering et al. (2017)). We used all retrievals for the year 2017 from version 8r that are flagged as "good" by this algorithm (O'Dell et al., 2012). Figure 4 shows that the physics has the strongest impact on the annual and seasonal climatology of x$CO_2$ fields. In boreal winter, the differences between the two physics exceed 0.5 ppm over tropical South America and tropical Southern Africa. In boreal summer, the differences are negative and exceed 0.3 ppm in terms of absolute value beyond 50° N. This is due to the weaker vertical mixing of the new physics which limits IH exchanges: the

negative anomalies of x$CO_2$ are more trapped into the northern hemisphere. Compared to the physics, the land surface scheme, the horizontal and vertical resolutions have a modest effect on x$CO_2$ with most differences less than 0.3 ppm. The values of 0.3 and 0.5 ppm mentioned here refer to, respectively, the threshold and breakthrough requirements for systematic errors in satellite retrievals as defined in the User Requirement Document of ESA's GreenHouse Gas Climate Change Initiative project (GHG-CCI, 2016). Comparing model performance to retrieval requirements is motivated by the same role that model errors

and retrieval errors play in atmospheric inversions. In our case, 6% of the winter land grid points and 5% of the summer land grid points in terms of the differences between the two physics exceed the 0.5-ppm minimum requirement.

If the horizontal resolution has a modest effect on the x$CO_2$ values at large scale, its impact can be much larger at local scale and exceed 0.5 ppm in individual grid points. The impact of the horizontal resolution is particularly noticeable over Northern India. In comparison, the effect of the land-surface scheme and of the vertical resolution are modest.

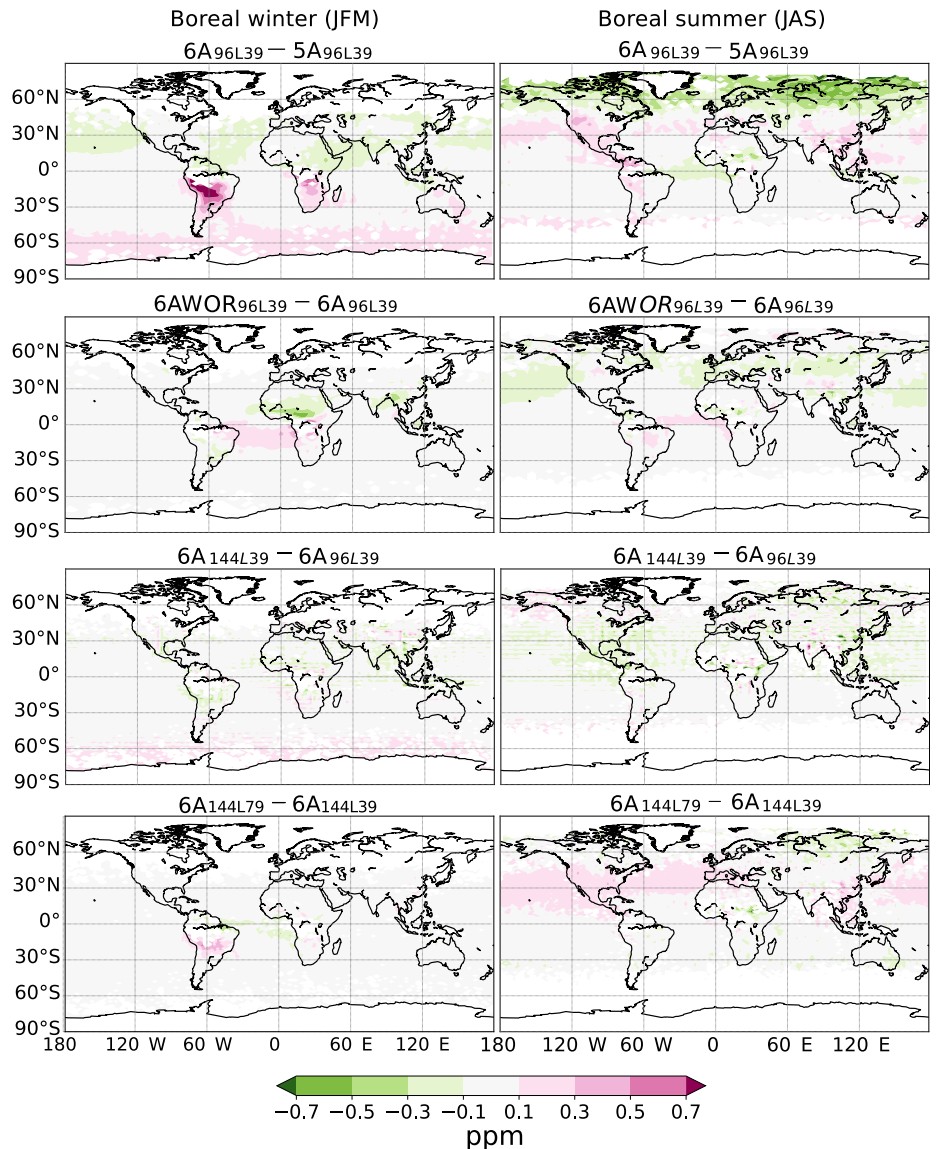

**Figure 4.** Map of the differences in $xCO_2$ (ppm) between 6A-96L39 and 5A-96L39 (top, effect of the new physics), 6A-96-L39 and 6AWOR-96L39 (second row, effect of the land surface), 6A-144L39 and 6A-96L39 (third row, effect of the horizontal resolution),6A-144L79 and 6A-144L39 (last row, effect of the vertical resolution). The left column shows the average over the 2005-2010 boreal winters (December-February) and the right column shows the average over 2005-2010 boreal summers (June-August). The simulated $xCO_2$ values have been temporally convolved with the sampling of the OCO-2 satellite retrievals for the year 2017.

## 4 Comparison with observations

### 4.1 Inter-hemispheric gradient with $SF_6$

A classical approach to evaluate the intensity of the IH exchanges is to plot the latitudinal distribution of the $SF_6$ mole fraction at the surface (Denning et al., 1999). The 5-year mean of the model-minus-observation mole fraction difference at the 11 background surface stations, in Figure 5, suggests that the IH exchanges are not sufficient in all versions as the gradient is systematically overestimated. The model spread has a value of 0.01 ppt for all latitudes, remaining smaller than the ensemble absolute bias of about 0.02 ppt. Both the ensemble spread and the ensemble bias remain usually smaller, by comparison, to the measurement calibration uncertainty of 0.03 ppt (96% confidence interval, NOAA ESRL GMD (2015)). The consistent negative difference of 0.01 ppt in the southern hemisphere induced by the new physics increases the surface latitudinal gradient and relates to the weaker vertical mixing. The vertical resolution cancels the effect of the physics by decreasing the latitudinal gradient and even improves it slightly.

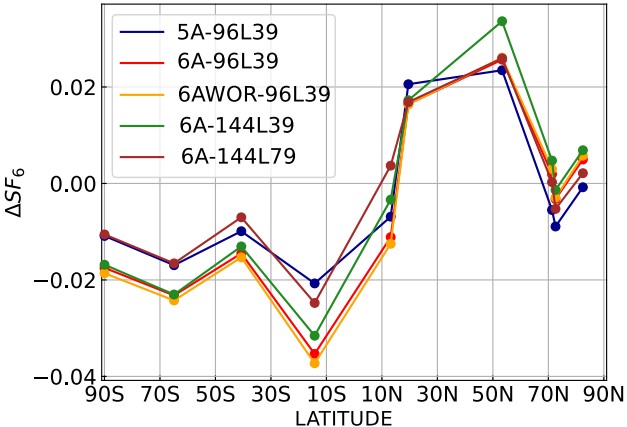

**Figure 5.** Latitudinal distribution of the $SF_6$ bias (modeled - observed) at the surface background stations during years 2005-2009. The stations used are: SPO, PSA, CGO, SMO, RPB, MLO, MHD, BRW, SUM, ALT.

### 4.2 Impact of the model setups on the $CO_2$ simulated concentrations at the surface and in the mid-troposphere

We now quantify the sensitivity of the simulated surface values of $CO_2$ to the model setups at annual, seasonal, synoptic and diurnal scales. From this perspective, we quantify the model spread of the simulated mole fraction for each surface site (total of 65 sites) that has been used for optimizing the prescribed fluxes during the years 1998-2014. Since the $CO_2$ fluxes have been estimated from a model version close to 5A-96L39, the best match between model and observations is expected to be obtained with this version. This is not necessarily true for the synoptic and diurnal scales which have not been constrained by inverse modelling. The location of the sites is depicted in Figure 1.

We also assess the ability of the different versions to represent unassimilated observations at surface sites located over the tropics. In the prescribed surface fluxes, the tropics represent $1.6 \pm 0.9 \, PgC/a$ of the $4.3 \, PgC/a$ global total flux averaged for years 2004-2011. Despite its importance, this part of the globe is not well constrained by inverse modelling systems (Peylin et al., 2013).

Last, we briefly look at the quality of the model simulations between 5 and 6 km above sea level by comparison to aircraft measurements. Aircraft measurements will be more extensively used in Section 4.3 in terms of profiles.

| VERSION | NORTHERN HEMISPHERE | TROPICS | SOUTHERN HEMISPHERE |
|---|---|---|---|
| 5A-96L39 | 1.7 (0.1) | - 0.4 (0.1) | -3.0 (0.1) |
| 6A-96L39 | 1.7 (0.1) | - 0.4 (0.1) | -2.8 (0.1) |
| 6AWOR-96-L39 | 1.5 (0.1) | - 0.2 (0.1) | -3.0 (0.1) |
| 6A-144L39 | 1.4 (0.1) | - 0.3 (0.1) | -3.0 (0.1) |
| 6A-144L79 | 1.6 (0.1) | -0.4 (0.1) | -2.8 (0.1) |

**Table 2.** Simulated mean gradients of $CO_2$ mixing ratios between MLO and other stations located in the Northern Hemisphere (latitudes $> 30°$ N), the Tropics ($30°$ S$\leq$ latitudes $\leq 30°$ N), the Southern Hemisphere (latitudes $< 30°$ S). For each one of the three domains, the corresponding sites are weighted by the inverse of their standard deviation. The value inside the brackets defines the associated mean weighted standard deviation.

### 4.2.1 Annual surface gradient to MLO

The annual gradient between stations reflects both large scale transport and integrated fluxes over large areas. Table 2 shows the mean and standard deviation of the annual gradient of the stations in the Northern Hemisphere, in the Tropics and in the

Southern Hemisphere, to MLO. On average over these latitudinal bands, the differences among simulations do not exceed 0.3 ppm and remain in the range of the measurement calibration objective defined by the WMO. 10 continental or coastal stations out of 65 assimilated surface sites (BRW, SHM, KAS, HUN, UTA, AMY, PAL, WLG, LEF, MHD) show differences larger than 0.3 ppm.

We performed the same analysis with a regional grouping of the stations, using the tiling of the globe in 22 regions defined

by the TransCom 3 protocol (Gurney et al., 2002). The largest systematic difference among the simulations is found for region Boreal North America (0.4 ppm), where the standard mean deviation around the annual mean is roughly 0.3 ppm for each simulation. In this case, Boreal North America is only represented by the inland site BRW which may not be representative of the whole region.

### 4.2.2 Seasonal variability

The impact of the model setups on the seasonal cycle at each station is documented considering two characteristics: the phase and the amplitude. The ratio of simulated to observed amplitude of the seasonal cycle is depicted for each station in the lower

panel of Figure 6 while the phase is displayed in the higher panel of Figure 6. For comparison purpose, the amplitude and phase are plotted separately for two versions simultaneously.

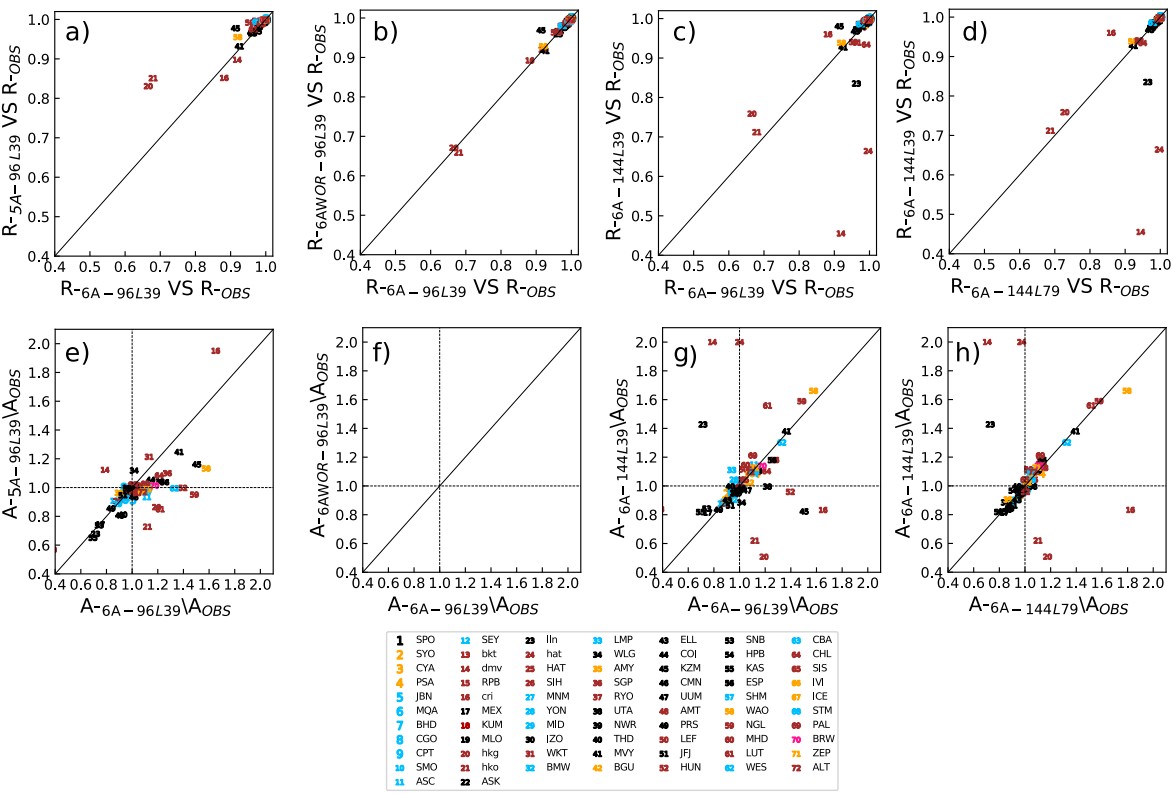

**Figure 6.** a) Correlations between the observed and simulated $CO_2$ mean seasonal cycles from 6A-96L39 (x axis) and 5A-96L39 (y axis) for all available stations. (b-d): Same as (a) but from versions 6AWOR-96L39, 6A-144L39, 6A-144L79. e) Ratio of the simulated to observed $CO_2$ seasonal amplitude from 6A-96L39 (x axis) and 5A-96L39 (y axis) for all available stations. (f): Same than a but from 6A-96L39 (x axis) and 6AWOR-96L39 (y axis). (g): Same than (f) but from 6A-96L39 (x axis) and 6A-144L39 (y axis). (h): Same than (f) but from 6A-144L39 (y axis) and 6A-144L79 (x axis). The stations are numbered by increasing latitude (with the identifier correspondence given in the bottom of the panel) and are colored according to their category. Blue: maritime stations, black: mountainous stations, yellow: coastal station, brown: continental station. Stations written in lowercase (uppercase) refer to unassimilated (assimilated) stations.

Regarding the phase (top row), most station points are located close to the bissector. This means that the phase is well captured (correlation above 0.9) and is not much affected by the model setups for most of the assimilated stations, including for station PSA (ratio of 1.1 and correlation of 0.99 with 6A-96L39) that was not well simulated by a previous intermediate version 6A (see Section 2.1). However, the seasonal features of the unassimilated stations (lln, hat, dmv, hkg, hko, cri) appear to be much more sensitive to the model setups, especially to the resolution. Station DMV is not depicted here since the correlation coefficient is less than 0.3 and the amplitude ranges between 0.3 and 0.6 depending on the model setup. The poor representation of the seasonal cycle of DVI has already been noticed by Lin et al. (2017). They attributed this deficiency to inaccurate prior

Net Ecosystem Exchange (NEE) and/or fire emissions in the prescribed surface fluxes as the $CH_4$ seasonal cycle was in better agreement with observations compared to the $CO_2$ simulated values in their model. This explanation is likely, given that the region is poorly constrained by observations. Because of their strong sensitivity to the model setups, these stations should be associated with a strong error if they are assimilated in the inverse system, which explains why they have been discarded so far

5   from the inversion system. The new physics increases the seasonal amplitude at (assimilated) mid-latitude sites over land: 9 stations out of 26 have an amplitude shift larger than or equal to 0.2 ppm as a result of the convective inhibition. The horizontal resolution has an impact limited to only 3 assimilated stations, that show an amplitude shift larger than or equal to 0.2 ppm. This is due to a change of topography and land fraction map. The amplitude at most mountain stations (7) is underestimated by more than 0.1 ppm in all versions even though they have been assimilated.

10      Figure 7 depicts the seasonal-mean latitudinal structure of the $CO_2$ bias (modeled - observed) at marine surface sites and at 5.5 km in boreal winter (JFM) and in boreal summer (JAS). In winter, the model spread reaches a value larger than 0.5 ppm both at the surface and at 5.5 km. In summer, the model spread reaches a value of 1.5 ppm near the surface beyond 40° N mainly due to the physics. Consistent with a less efficient mixing inferred in the zonal mean structure (Figure 3), the new physics decreases (increases) the latitudinal gradient in boreal mid-latitudes in summer at the surface (at 5.5 km) as the negative anomalies are

15   more trapped in the boundary layer. For all simulations, the latitudinal gradient at 5.5 km between 50° N and 40° S is well reproduced as the bias does not exceed 0.5 ppm.

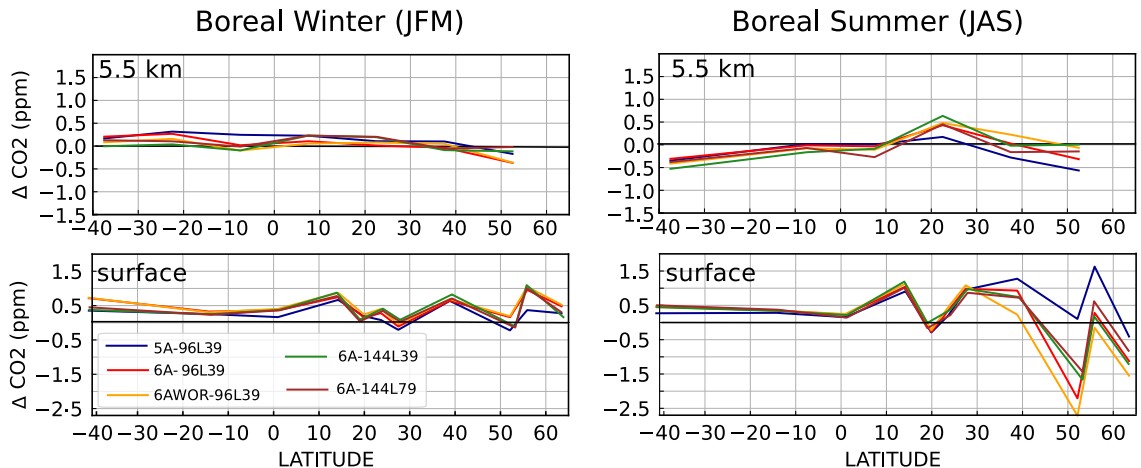

**Figure 7.** Latitudinal mean distribution of the $CO_2$ bias (modeled - observed) between 5 and 6 km above sea level in the free troposphere (upper) and at the marine boundary layer (MBL) sites (lower) for January-February-March (JFM) (left) and July-August-September (JAS) (right) during the period 2007-2010. The MBL sites are ZEP, ICE, SHM, AZR, MID, MNM, KUM, GMI, SMO, CGO. The 5-6 km measurements come from the CONTRAIL database.

### 4.2.3 Synoptic variability at the surface

The synoptic variability characteristics, Normalized Standard Deviation (NSD) and correlations with observations, are depicted for each station on a Taylor Diagram in Figure 8. NSD refers to the ratio of the simulated to observed standard deviation. Consistent with the design of Taylor diagrams, the distance between an actual model result and the reference (the star) is equal to the relative root mean square error (RMS). Unsurprisingly, the model-minus-observation mismatch is not as good as for the seasonal variability. Indeed, the synoptic scale has not been constrained by the inverse modelling system. In the reference version, most stations (58 out of 72) have correlations around 0.8 and a NSD around 0.7. The lack of synoptic variability in 5A-96L39 has been reported over Europe (Locatelli et al., 2013) and over Asia (Lin et al., 2017). All versions of the model have difficulties in accurately reproducing the synoptic variability at the mountain stations. The new physics enhances the standard deviation at some sites located in the northern mid-latitudes. The horizontal resolution has a mixed impact: It slightly increases the amplitude but increases or decreases the correlation coefficient depending on the sites. This can be attributed to the coarse resolution of the prescribed fluxes or to NWP forcing uncertainties. The synoptic variability is not affected by the land surface scheme nor by the vertical resolution. As for the seasonal variability, the improved horizontal resolution has a limited impact on the simulated synoptic variability to only 3 assimilated sites (KZM: 45, CHL: 64, HUN: 52) in terms of amplitude and correlations with observations. All versions poorly simulate synoptic variability at site hko since the site is located in an urban area and is affected by local emissions not well described in the prescribed surface fluxes.

### 4.2.4 Diurnal cycle at the surface

The simulated $CO_2$ diurnal variation reflects the day-night contrast in both the prescribed fluxes and the PBL (Planetary Boundary Layer) vertical mixing. Since the fossil fuel emission inventory is here constant within a month, most of the diurnal variability comes from the prior biospheric fluxes, with marginal corrections having been brought by the inverse modelling system. Another part of the diurnal variability is induced by boundary layer processes: during night-time, $CO_2$ accumulates near the surface within the shallower stable boundary layer whereas during daytime, the low $CO_2$ concentration caused by the photosynthesis uptake is distributed over a deeper convective PBL. The daily-mean $CO_2$ mole fraction would be positive even when the integrated flux over the day is zero (Denning et al., 1995). This diurnal rectification highlights the importance of diurnal cycle representation, since its lack of realism might have repercussions on longer timescales.

Figure 9 shows the peak-to-peak amplitude of the $CO_2$ mole fractions for 8 sites with an amplitude greater than 1.5 ppm for the boreal summer months (JJA). Although similar conclusions can be drawn in boreal winter, we depict diurnal cycle characteristics only for the summer when the diurnal amplitude is the strongest. We can see that for most sites, version 5A underestimates the diurnal amplitude with the exception of AMY, in agreement with previous studies (Geels et al., 2007; Locatelli et al., 2015a). The new physics increases the diurnal amplitude at continental sites AMY, MVY and NGL, especially regarding the extremes. Locatelli et al. (2015a) in their supplement showed that the Mellor and Yamada (1974) scheme strongly increases $^{222}Rn$ overnight compared to the Louis (1979) scheme used in the 5A version. Similar experiments with $^{222}Rn$ lead to the same conclusion (not shown). The strongest increase of amplitude (up to 10 ppm) is seen with a finer vertical resolution

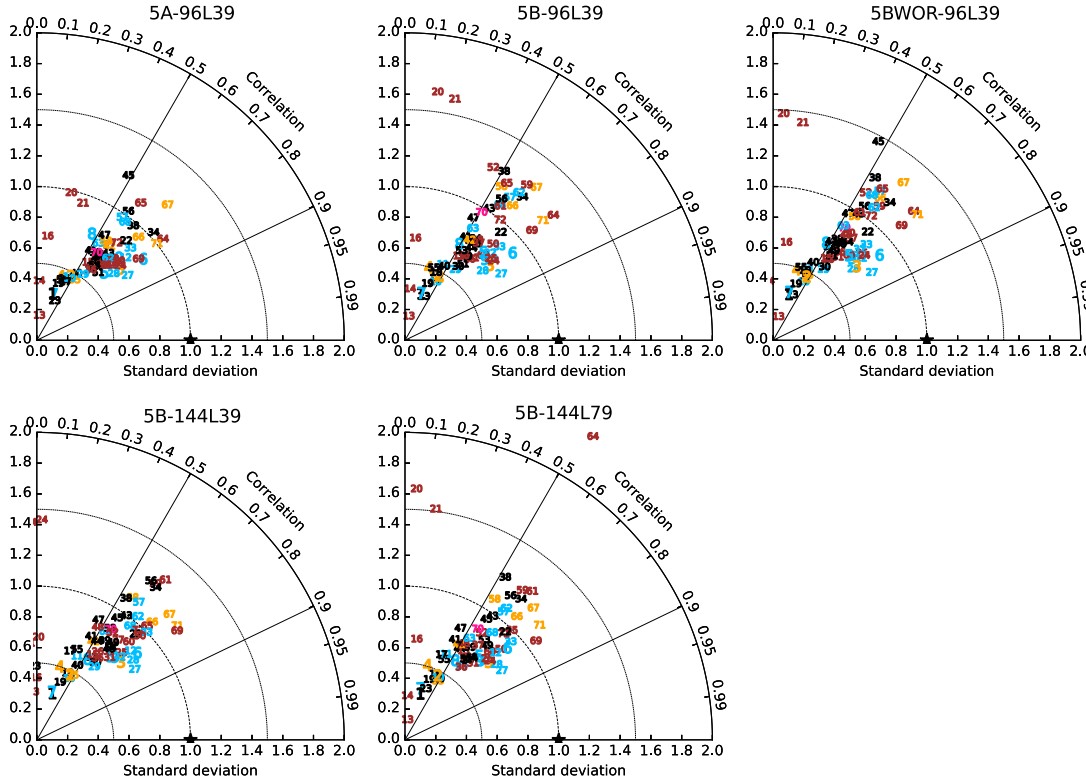

**Figure 8.** Taylor diagrams showing correlations and normalized standard deviations (NSD: the ratio of the simulated to observed standard deviation) between the simulated and observed $CO_2$ synoptic variability for all surface stations. The stations are numbered and coloured as in Figure 6.

for the continental stations NGL and AMY. A possible explanation is that the $CO_2$ input from the surface is distributed within a thinner layer. The lower panel of Figure 9 shows boxplots of a measure of the phase of the diurnal cycle at the same sites in boreal summer for the $CO_2$ simulated mole fraction and the $CO_2$ prescribed fluxes. The measure of the phase is defined as the local time at the minimum $CO_2$ mole fraction. It typically happens in the afternoon after the convection has ventilated the

5    PBL and the photosynthesis activity has drained the $CO_2$ at the surface. In the GCM, the minimum value of the fluxes to the atmosphere seems to propagate to the sampling level within a few hours at each site. The new physics affects the amplitude without noticeably ameliorating the timing of the diurnal cycle. The timing at mountain site SNB is improved whereas it is deteriorated at site PAL (516 m). The other sites are not affected by the change of physics. In contrast, the horizontal resolution seems to have a positive effect both on the timing and the amplitude at coastal site MHD. All versions seem to

10   underestimate the mean amplitude and shift the daytime minimum earlier at the mountain sites CMN and bkt compared to lower-latitude sites. Nonetheless, the amplitude is largely dependent on the sampling location and model level. Models typically

show high amplitudes at model levels close to the surface and smaller amplitudes aloft (Law et al., 2008). In order to improve the representation of the diurnal cycle, it might be preferable to choose the level which better fits the observations.

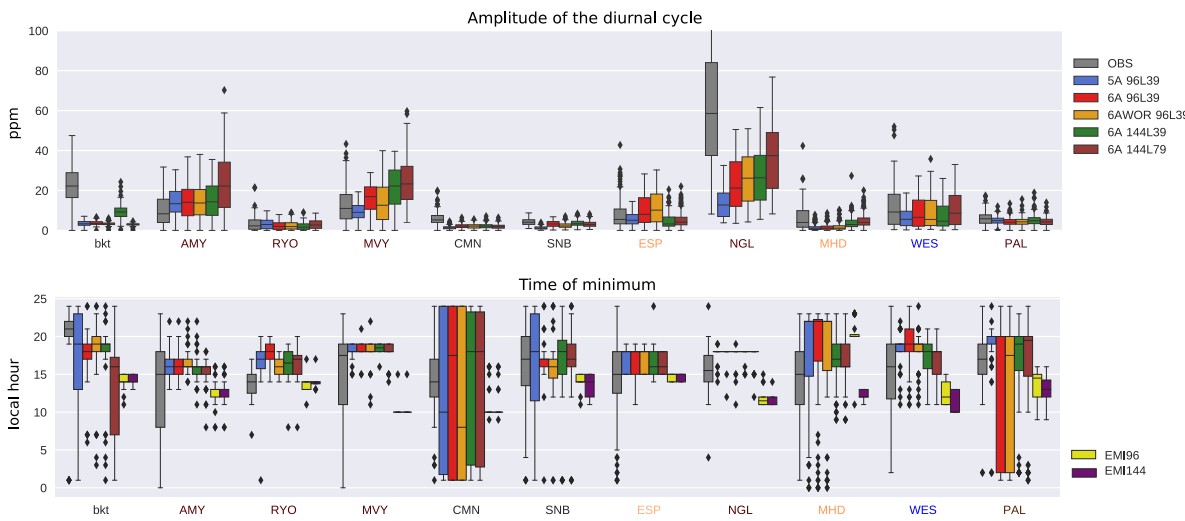

**Figure 9.** Top: Boxplots of the peak-to-peak amplitude (maximum concentration minus minimum concentration) of the mean diurnal cycle for July-September for observed (grey) and modelled (colors) $CO_2$ for each model simulation during the years 2011-2012. The diurnal amplitude is calculated from the residual between the raw data and the daily mean. The sites are listed on the abscissa. Bottom: Boxplots of time of minimum crossing for each model. The time for the prescribed $CO_2$ are displayed for both horizontal resolutions in yellow ($96 \times 95$) and purple ($144 \times 143$). Here are depicted only the sites with a diurnal amplitude greater than 1 ppm. The code color for stations is the same as previously.

## 4.3 Validation against independent measurements of vertical profiles of $CO_2$

Errors on $CO_2$ flux estimates by inverse modelling are thought to be proportional to the vertical mixing efficiency within a column (Stephens et al., 2007; Saito et al., 2013). If a model transports too much tracer from the boundary layer to the free atmosphere, the inverse system will compensate the induced tracer deficit at the surface by modulating the $CO_2$ fluxes. A mean of validating the flux estimates is to compare the simulated vertical profiles with independent (unassimilated) observations of vertical profiles (Pickett-Heaps et al., 2011). Since only surface measurements have been assimilated, the vertical gradient mainly reflects intrinsic mixing efficiency within the column. In this section, we evaluate the simulated vertical profile against independent aircraft measurements over several regions: Europe, North America, Brazil, East Asia, Greater Northern India, Northern Southeast Asia at the annual and seasonal scales. The benefit of using the newly developed version is also assessed over these regions.

### 4.3.1 North America and Europe

Over North America, the surface flux pattern has a strong seasonality. In winter, positive fluxes to the atmosphere driven by fossil fuel emissions are mainly located along the East coast whereas in summer, the strongest sink is located over the mid-West states. Because of a large net ecosystem production (NEP) of organic carbon during the crop plant growth, the mid-West states can contribute to half of the summer uptake in North America (Crevoisier et al., 2010; Sweeney et al., 2015). $CO_2$ fluxes over North America are relatively well constrained by surface observations as seen in Figure 1.

Figure 10 shows the seasonal and annual climatologies of the $CO_2$ mole fraction bias (model-observations) on average over all the North American airborne platforms depicted in Figure 1. On the whole, the simulated value in the lowest level is overestimated by about 0.5 ppm on an annual basis and by about 1 ppm in winter. This behaviour is seen both for profile sites close to assimilated stations (ESP, LEF, THD, SGP) and for profile sites further away (not shown). The profile above 2 km is well simulated except in summer when the bias is about 0.5 ppm. This leads to an overestimated vertical gradient between 1 and 3 km in winter. In the inversion system, the overestimated winter gradient would artificially decrease the estimated fluxes to the atmosphere. The model spread does not exceed 0.5 ppm throughout the year except in summer when it reaches a value of 1.5 ppm at 1.5 km and 1 ppm at ground level. It only explains a small share of the variability (standard deviation) of the differences (about 1-1.5 ppm). This variability of the differences is comparable among the model versions. The difference between the two physics is responsible for a large portion of the model spread. This can be explained, in part, by the fact that the air mass composition is more influenced by local processes during the summer than at any time of the year. At each site, westerly wind flow prevails throughout the year in the entire free troposphere. As the air masses move across the continent, they progressively mix with air coming from the biosphere and from fossil emissions. In summer, the decrease of the wind speed over the mid-continent and over the East coast results in less homogeneous vertical profiles in the free troposphere (Sweeney et al., 2015). Combined with enhanced convection, this effect might emphasize the divergence between the two physics. The convective inhibition (Figure 3) as a result of the new physics translates into a lower concentration of 1 ppm at 1.5 km and a higher $CO_2$ concentration of 0.6 ppm in the mid-troposphere as the trapping of negative anomalies of the $CO_2$ mole fraction within the PBL is enhanced. The $CO_2$ depletion around 1.5 km induced by the new physics may be due to the vertical transport of negative anomalies by the thermal activity. Combined with the new physics, the land surface scheme also has a strong impact on the summer vertical profile as the amount of water vapour and temperature directly influence the vertical mixing through surface buoyancy. By inhibiting deep convection, it increases the upper troposphere concentration by 0.5 ppm and decreases the surface concentration by 0.5 ppm. The effect of the resolution is modest here.

The figure for Europe (EUR, Figure 11) shows similar features as for North America, but with smaller model-measurement differences (absolute biases, standard deviations, model spread), except for the standard deviations in the lower atmosphere that are about 50% larger.

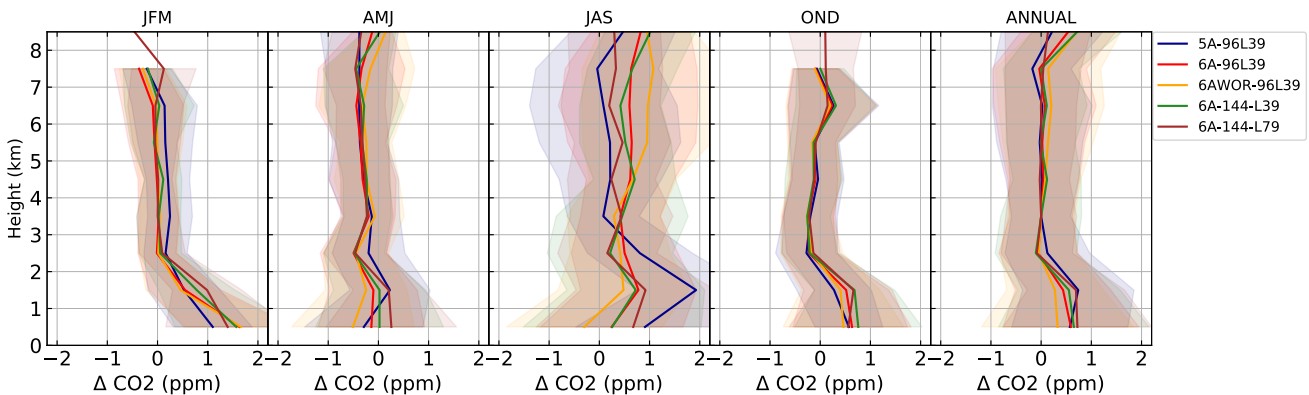

**Figure 10.** Bias (model-observations, thick lines) and associated standard deviation (shaded areas) for the monthly $CO_2$ vertical profile differences over North America during the period 2008-2014. The data have first been averaged in 1-km-altitude-bins per hour and per site, before being averaged among the 12 North American sites of Figure 1 per month. The statistics are drawn from that ensemble of monthly and spatially-averaged values. They are shown for each season (January-March, JFM; April-June, AMJ; July-September, JAS; October-December, OND) and for the whole year. In order to highlight the differences in profile shape, the annual mean of the bias at 3.5 km has been removed for each simulated vertical profile (5A-96L39: -2.0 ppm, 6A-96L39: -2.0 ppm, 6AWOR-96L39: -2.0 ppm, 6A-144L39 and 6A-144L79: 1.3 ppm).

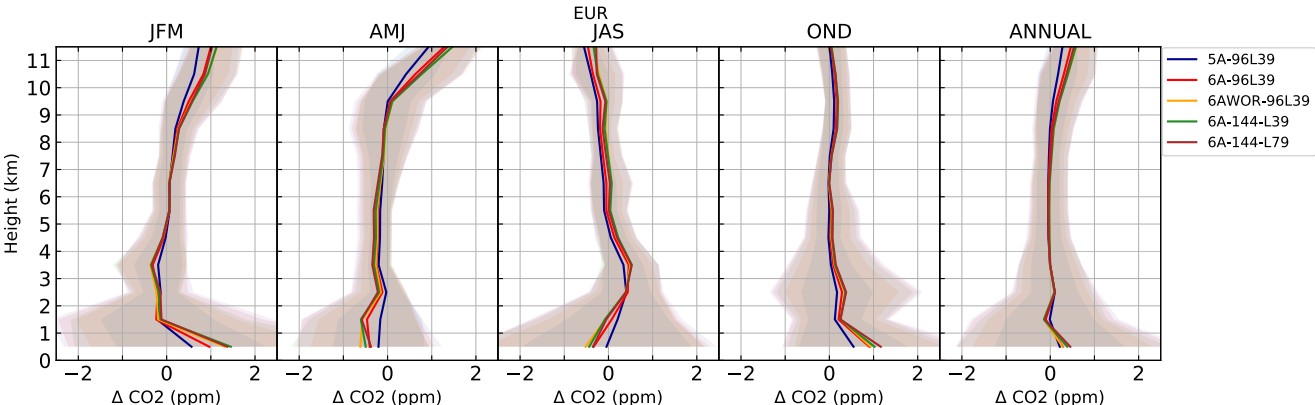

**Figure 11.** Same as for Figure 10 but over Europe from the CONTRAIL dataset during the period 2006-2011. The domain is portraited in Figure 2.

### 4.3.2 Indo-Pacific region

Figure 12 presents the profile of the model-observations difference statistics for the CONTRAIL $CO_2$ data over Eastern Asia, Northern Southern Asia, and Greater Northern India. They mostly have the same shape: a negative bias close to the surface (up to -8 ppm for Greater Northern India in OND) and a null one above. The decreasing standard deviations of the differences with height and the small model spread under 1 ppm are similar to EUR, except for Greater Northern India in the lower atmosphere,

where the model spread reaches 2 ppm (up to 4 ppm) at the seasonal scale, in particular at the end of the monsoon season (OND).

In NSA and IND, the negative bias at annual scale within the boundary layer is likely related to urban sources, close to the airports for these commercial flights. The negative bias was also noticed in NSA and in IND for OND in the study of Lin et al. (2017). We also note that the prescribed surface fluxes have not been well constrained for IND and NSA. For NSA, only station GMI has been assimilated over the period 2006-2011. IND is directly constrained by HLE only, a site that is located in a mountain area at the northern edge of the domain: backward trajectories showed that the site HLE samples air masses moving through the Arabian desert and North Africa in winter and those coming from Southeast Asia in summer (Suresh Babu et al., 2011; Lin et al., 2015). The impact of the model setups reaches 3 ppm in this region and during AMJ and OND, two intermediary seasons. Special care should be taken when assimilating new stations in this area. Further to this lack of measurement constraints, the prescribed flux variability in NSA and IND mainly reflects the prior flux variability, while in EAS, fluxes are more robust (Thompson et al., 2016) and the model-observations differences appear comparable to EUR there.

### 4.3.3 Amazon basin

The $CO_2$ surface fluxes over the Amazon basin have not been directly constrained by observations. The two closest assimilated stations are located along the Atlantic coast (Figure 1). They are representative of the air masses coming off the tropical Atlantic ocean through the tropical easterly winds (Gatti et al., 2014). Moreover, the assimilation of additional surface and airborne observations has not enabled the variability of the $CO_2$ fluxes to be improved so far, at least with this inversion system (Molina et al., 2015). Molina et al. (2015) concluded, through several experiments with both global and regional models, that this limitation mainly stems from model transport errors and uncertainties on biospheric and fire burning emissions. In this context, we evaluate the sensitivity of the simulated $CO_2$ concentrations to model setups at the four airborne stations featured in Figure 1: tab, rba, alf, san. The simulated and observed $CO_2$ vertical profile averaged for the wet period (January-June) and dry period (July-October) in 2010 are depicted in Figure 13. All versions poorly represent the shape of the mean observed $CO_2$ vertical profiles in the lower troposphere. The mismatch is particularly amplified during the dry season. The vertical gradients of the reference 5A-96L39 and of the observations between 1 km and 3 km have opposite signs, suggesting issues in the prior fluxes (NEE or/and fire emissions).

The simulated profile is also very sensitive to the subgrid scale parameterizations for each site, and, to a lesser extent, to the land surface model. At the surface, the differences between the two physics ranges from 2 ppm at san in the dry season to 6 ppm at tab during the wet period. The other setups have a modest impact compared to the physics.

The $CO_2$ vertical profiles suggest a more mixed lower and mid troposphere with the new physics. In order to visualize the behaviour of the two physics, we additionally calculate the corresponding simulated $^{222}Rn$ profiles with the same sampling strategy, even though we do not have any observations to compare them with. The lower panel of Figure 13 shows that less radon is transported above 5 km in the new physics, suggesting a less dominant role of deep convection. This is confirmed while comparing the simulated mean precipitation during the wet and dry period with reference data from NASA's Global Precipitation Climatology Project (Figure 14). In the Tropics, precipitation is an indicator of the convective activity and we

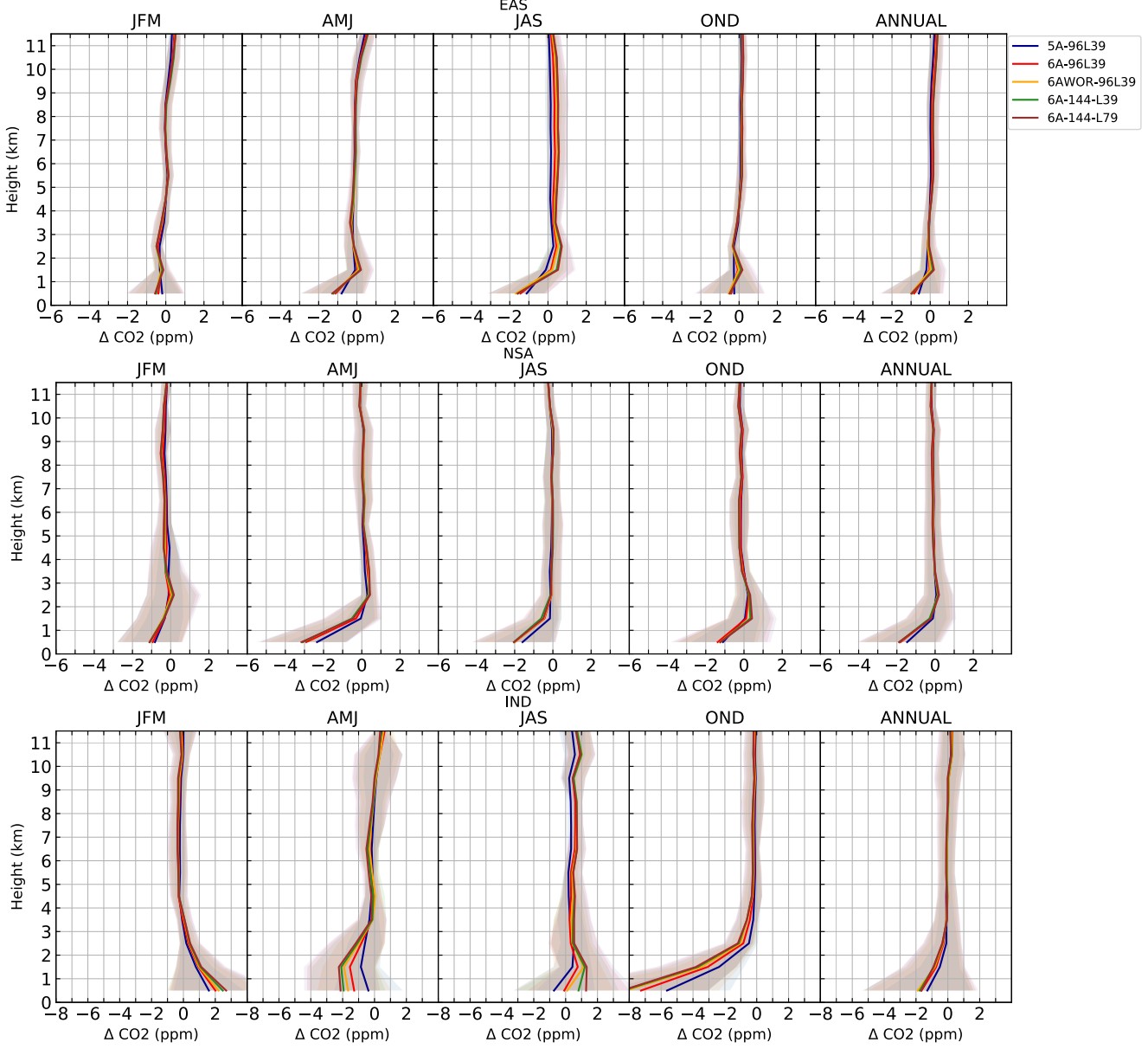

**Figure 12.** Same as for Figure 11 over Eastern Asia (EAS), Northern Southeast Asia (NSA) and Greater Northern India (IND).

see here that the new physics decreases the mean precipitation (mainly convective) during both periods without showing better agreement with the reference data. The modelling of the precipitation in this region has been shown to be particularly challenging (Lintner et al., 2017). The simulated radon profiles suggest that more radon is detrained above the boundary layers by the thermals in the new physics, especially during the dry season. The strengthening of the thermals when the deep

convective scheme is inhibited is a known behavior of the new physics (Rio and Hourdin, 2008). As a result, the boundary layer of the new physics is more mixed and goes higher.

The lack of realism of the simulated local transport does not impact the $CO_2$ fluxes estimated by inverse modelling in this region, as they mostly rely on the prior fluxes and long-range transport up to now. However, it limits the potential benefit of assimilating new surface observations there, in line with Molina et al. (2015).

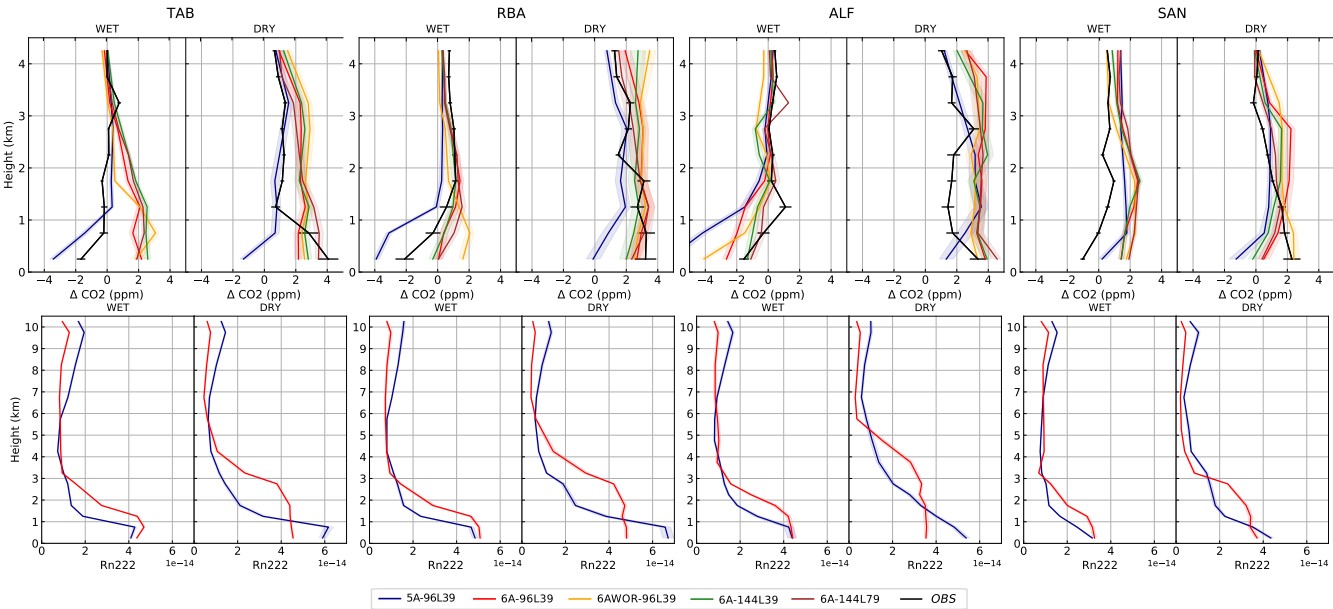

**Figure 13.** Top: Mean difference between $CO_2$ profiles measured and simulated in 2010 at the four Amazonian aircraft sampling sites and an oceanic $CO_2$ background (that is, $\Delta\ CO_2$) during the dry (left of each panel) and wet (right of each panel) seasons, respectively (solid lines) and the standard deviation divided by the square root of number of profiles (dashed lines and error bars). The background is estimated from in situ measurements at monitoring stations ASC and RPB, as described in the main text. Bottom: Same as the top but for the $^{222}Rn$ (ppm). The dry season (red lines) is affected by fires at most sites and is here defined as period July-October for illustrative purposes only; it does not correspond to all months within the fire season.

## 5 Conclusions

We have compared two reference versions of a GCM, LMDz, that have been prepared for, respectively, CMIP3 and CMIP6, from the point of view of the transport of tracers. The more recent version benefits from a more elaborated radiative scheme and subgrid scale parameterizations, in addition to a refined vertical resolution. The main changes in the physical parameterizations concern boundary layer mixing due to vertical diffusion (Mellor and Yamada, 1974), shallow convection (Rio and Hourdin, 2008; Rio et al., 2009), thermodynamic effects of ice, cool pools (Grandpeix et al., 2009) and convective triggering and closure assumptions (Rochetin et al., 2013). These main changes have been accompanied over the years by other evolutions of the

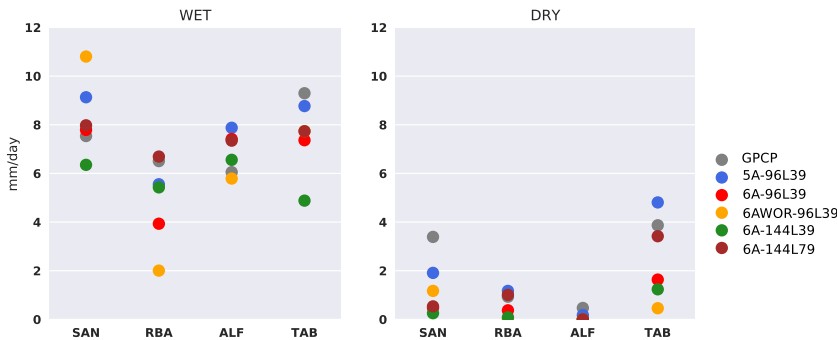

**Figure 14.** Observed and simulated mean precipitation (mm/day) during the wet and dry seasons over each Amazonian sampling site (tab, rba, san, alf). The black dots depict monthly mean precipitations derived from NASA's Global Precipitation Climatology Project.

model physics, by continuous tuning (Hourdin et al., 2017), and by continuous technical changes (including bug introduction and bug fixes) that have diverse impacts. Within this flow of modifications from a large developer group, our evaluation of the two versions is based on a snapshot of the LMDz code in its release 2791, a few months before the start of the CMIP6 simulations.

We performed a set of $CO_2$, $SF_6$ and $^{222}Rn$ simulations using those two versions of LMDz at two horizontal resolutions and guided by the ECMWF wind reanalysis for nearly two decades (1998-2014). In addition, we compared two simulations with two different land surface schemes, one using the ORCHIDEE terrestrial surface model and the second using a simplified bulk scheme. In this case, the land surface scheme only controls the heat and latent fluxes at the land-atmosphere interface. The $SF_6$ and $^{222}Rn$ emissions were prescribed following the TransCom 3 protocol. The $CO_2$ surface fluxes have been optimized

beforehand by the assimilation of surface observations in a version of LMDz close to the older model version studied here. We have compared the resulting ensemble of simulations with both assimilated and unassimilated $CO_2$ observations from a large dataset in different parts of the globe. This study enabled us to benchmark the effects of the resolution, land-surface scheme and sub-grid scale parameterizations on $CO_2$ simulated values, which is a fundamental step before implementing the recent developments in our inverse modelling system.

At the surface, the comparison with the assimilated $CO_2$ measurements showed that the land-surface scheme and the vertical resolution have a limited impact compared to the horizontal resolution and sub-grid scale parameterizations. The new physics tends to weaken the vertical mixing within the column over continental areas. The annual mean mole fraction values are little modified but the variability at seasonal, synoptic and diurnal scales is enhanced at continental and coastal sites. The higher seasonal cycle in the northern hemisphere, as a result of a less efficient vertical mixing, affects the latitudinal $CO_2$

gradient in boreal summer by about 1 ppm, a value that should impact the geographical distribution of the $CO_2$ surface fluxes estimated by inverse modelling. At synoptic scale, the higher model variance does not lead to an improved correlation with the measurements: the change of physics increases the amplitude of the synoptic variability without affecting the phasing. As for the diurnal cycle, even though the amplitude shows better agreement with the observations, the phasing is not improved by

the model setups at most $CO_2$ monitoring stations, but it heavily relies on the prior fluxes used in the inversion system. Even though the improved amplitude is promising for assimilating a larger fraction of hourly data at continental surface stations, further efforts should be made on the prior biospheric fluxes and on sub-grid scale parameterizations to better simulate the diurnal cycle. The atmospheric transport at mountain stations is still poorly captured by all versions even when considering the refined vertical and horizontal resolutions. This may mean that the resolution is still too coarse to accurately reproduce the atmospheric flow around complex topography. The annual-mean latitudinal gradient of $SF_6$ is still slightly too strong in all model versions, likely reflecting insufficient IH exchanges.

The assimilation of column-average mole fraction retrievals from satellites like OCO-2 offers a promising perspective for atmospheric inversion because their spatial density joined to their vertical integration reduce the impact of transport errors (Basu et al., 2018). From that perspective, we quantified the impact of the model setups on the simulated x$CO_2$ convolved with the OCO-2 space-time coverage for a given year. The model-ensemble spread is mainly due to the physics and exceeds 0.5 ppm in the boreal summer high latitudes, in Northern Africa and in Brazil, or locally around emission hot-spots. In boreal summer, the new physics decreases the latitudinal gradient by decreasing the x$CO_2$ values in the high latitudes further due to less efficient vertical mixing. This may decrease the northern sink inferred by inverse modelling with the satellite data and LMDz. In austral summer, the mean x$CO_2$ shows large discrepancies (up to 1 ppm) over the Amazonian basin between the simulations, this region being particularly sensitive to the parameterization assumptions. As for the surface fields, the x$CO_2$ fields are sensitive to the horizontal resolutions around emission hot-spots.

The comparison with unassimilated airborne measurements enabled assessment of the quality of the inversion as well as the sensitivity of the vertical profiles of $CO_2$ mixing ratio to the model setups. The results show that the accuracy of the simulated $CO_2$ vertical profiles as well as their sensitivity to parameterizations depend on the region of interest and the season. Profiles in regions well covered by observations such as Europe, Eastern Asia and North America tend to be better captured than in regions poorly constrained (Greater Northern India, Northern Southeast Asia, Brazil). The optimized fluxes reflect mainly the prior fluxes in these regions nearly devoid of assimilated data. Over the Amazonian basin, the present study indicates that the vertical profile uncertainty mainly comes from the physical parameterizations and, to a less extent from the land surface model, with a model spread reaching 6 ppm in the boundary layer. Here again, a finer resolution does not noticeably modify the vertical profile shape. This does of course underline the large uncertainties associated with the optimized fluxes and the difficulties in assimilating new observations over these regions, confirming the findings of Molina et al. (2015). Given the leading role of the Amazonian basin on the global carbon cycle, it appears important to improve the realism of the vertical mixing over this region. For example, radon profile samples operated by airborne campaign could help the modelling community to improve convective parameterizations at specific sites.

In terms of CPU time, the most advanced version tested here (6A-144L79) is about twenty times more expensive than the reference version (5A-96L39), due to refined spatial and temporal resolutions, and to more sophisticated sub-models. If adapted to the off-line configuration that is used in the atmospheric inversion system, it would be at least five times more expensive than the current version due to the refined horizontal and vertical grids, but the time step may also have to be reduced for the whole code and even much more within an off-line version of the new thermal plume model. It will be possible to distribute

the computational load on a large number of processing units with the new icosahedral dynamical core of LMDz, when it has been coupled to the LMDz physical package and then adapted to the off-line model (Dubos et al., 2015). In the mean time, we may wonder if the benefit of the new version for $CO_2$ atmospheric inversion counterbalances its numerical expense. To address this question, the sensitivity of the $CO_2$ surface values to the model setups gives some insights into their impact on the inferred surface fluxes. On a seasonal basis, the updated physics would likely decrease the northern sink in boreal summer as a result of a weaker vertical mixing within the column. However, the robustness of the simulated surface concentration gradients to MLO suggests that, on an annual basis, the large-scale surface fluxes inferred from surface measurements using an updated version should remain the same, meaning that the increased boreal summer uptake would be compensated in the rest of the year. Further, LMDz versions developed in the last few months, after the ones tested here, appear to strengthen vertical mixing within the column again (results not shown). In this context, only the horizontal resolution is expected to bring some improvement on the estimated natural fluxes depending on the quality of the prior fossil fuel emission inventory. However, when assimilating satellite observations, annual-mean flux estimates in the high latitudes should change because of the interaction between the changed flux seasonal cycle and the seasonally-varying satellite sampling (Byrne et al., 2017). The improved vertical resolution from 39 to 79 layers has a marginal impact on the simulated $CO_2$ values, a situation which is different from the previous change from 19 layers to 39 layers, that had a major benefit in the inversion system (Locatelli et al., 2015b).

Even in the cases when the model setups have significant impact, our experiments, that are classical in the TransCom community, did not allow us to demonstrate the superiority of one version versus another. All versions appear to represent valid transport modelling options (at least with the current data selection in the inversion system) and the motivation to implement the most sophisticated one in the inversion system would mainly come from the wish not to diverge from the LMDz core technical and scientific developments. This situation is paradoxical given the major improvements brought to LMDz for the representation of meteorology and climate, the benefit of which on other variables than tracer concentrations can be seen even when horizontal winds are nudged, like here (Hourdin et al., 2015). However, we may miss some measurement programs dedicated to the transport of tracers in the column. Observations of mixing boundary layer heights from radiosondes, ceilometers or satellites may also give some insight into the model quality, as well as the comparison with some highly detailed models (e.g., Randall et al., 2003).

*Code and data availability.* The LMDz model is available from http://web.lmd.jussieu.fr/trac under the CeCILL v2 Free Software License.

*Competing interests.* The authors declare no competing interest

*Acknowledgements.* The authors thank the LMDz developers for maintaining the dynamism and creativity of this climate model. The present work was granted access to the HPC resources of TGCC under the allocation A0030102201. It was funded by the Copernicus Atmosphere Monitoring Service, implemented by the European Centre for Medium-Range Weather Forecasts (ECMWF) on behalf of the European Commission. We also gratefully acknowledge the many people who contributed atmospheric observations (aircraft and surface air sample measurements). OCO-2 retrievals were produced by the OCO-2 project at the Jet Propulsion Laboratory, California Institute of Technology, and obtained from the OCO-2 data archive maintained at the NASA Goddard Earth Science Data and Information Services Center. We also thank Martin Steinbacher, Luciana Gatti, Antoine Berchet and Matthew Lang for their comments on an earlier version of the text.

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
