# Peer review of "On the impact of recent developments of the LMDz atmospheric general circulation model on the simulation of $CO_2$ transport"

_Geoscientific Model Development, 2018_

## Short Comment (SC1) · 9 Jul 2018

Dear authors,

in my role as Executive editor of GMD, I would like to bring to your attention our Editorial version 1.1:

http://www.geosci-model-dev.net/8/3487/2015/gmd-8-3487-2015.html

This highlights some requirements of papers published in GMD, which is also available on the GMD website in the 'Manuscript Types' section:

http://www.geoscientific-model-development.net/submission/manuscript_types.html

In particular, please note that for your paper, the following requirements have not been met in the Discussions paper:

- "The main paper must give the model name and version number (or other unique identifier) in the title."

- "If the model development relates to a single model then the model name and the version number must be included in the title of the paper. If the main intention of an article is to make a general (i.e. model independent) statement about the usefulness of a new development, but the usefulness is shown with the help of one specific model, the model name and version number must be stated in the title. The title could have a form such as, "Title outlining amazing generic advance: a case study with Model XXX (version Y)"."

As your article compares different LMDz versions please add the model name to the title in your revised submission to GMD, e.g., "On the impact of recent developments of an atmospheric general circulation model on the simulation of CO2 transport: a case study with LMDz"

Yours,

Astrid Kerkweg

---

## Referee Comment (RC1) · C.D. Jones (Referee) · 29 Jul 2018

Review of "On the impact of recent developments of an atmospheric general circulation model on the simulation of CO2 transport", by M. Remaud et al.

As ESMs are more frequently used for climate projection and many of them include explicit simulation of CO2 transport and its 3D structure in the atmosphere, it is clearly becoming more important to adequately evaluate whether the dynamics and transport within these models is fit for purpose, and to understand the implications of any model errors or biases.

Over recent years, there has not been widespread interaction between the carbon cycle/GCM/ESM community (such as C4MIP) and the trace-gas transport community such as Transcom. This paper is therefore timely and attempts to address this lack with a detailed examination of the performance of tracer transport in the LMDz GCM. The model has been developed for use in CMIP6 and although not explicitly mentioned, presumably aims to perform the C4MIP CO2-emissions-driven simulations. Its behaviour is compared against an older version, and the impacts of several developments are assessed in some detail.

I am not an expert in the details of transport schemes or their evaluation, but review this in the context of its potential usefulness for CMIP6/C4MIP climate simulations. I found the manuscript well written and clearly structured. The experiments performed are logical, well described and analysed logically and clearly.

Overall I find the manuscript well suited to publication in GMD and recommend mainly minor amendments.

I have one specific query though. I am concerned by the apparent circularity of deriving surface CO2 fluxes using LMDx in an inversion system and then using these fluxes as input in a forward transport experiment to test against observations. Surely this is more than the passing mention in section 2.3, "... the surface fluxes carry some imprint of a ... [LMDz]...". You mention that other inversion products exist and could have been used - so why did you choose one so closely related to the model you were evaluating? At the very least I would like to see some defence of this choice - can you show and quantify the different sources available? How different are they? Some of your intro text describes how surface CO2 fluxes are still very ill constrained in many places, so I imagine the different sources available to you may well be quite different. I could envisage some value in using multiple (or at least 2) sources and showing an additional panel to figure 3 - how does the choice of surface fluxes compare with changes in resolution and surface physics?

A few minor points are listed below, but if you can address this main concern then I would consider this a very sound paper.

Chris Jones

1. section 2.2. finishes by mentioning a mass correction - can you elaborate slightly? Is this simply a diagnoses of the loss on each tilmestep which is then added back evenly distributed through space?

2. in a couple of places you mention CMIP4. I am assuming this is meant to say CMIP5? (but could also mean CMIP3 which accompanied AR4, or it could mean C4MIP - the confusion of MIPs is endless!). Please check and clarify.

3. section 4.3 uses independent measurements of vertical profiles - this seems like a more reasonable (i.e. not circular on the choice of transport model used for the inversion that produced the surface fluxes) test. If this is really the case then I'd expect the results here to be (almost) independent of your choice of surface fluxes, so that would be a good check if you performed a run with different sources - do the results in this section still come out the same?

4. related to my main point, I'm quite happy with conclusions about the impact of processes such as "the new physics tends to weaken the vertical mixing" - this seems a reasonable conclusion from your analysis. Where I am less comfortable are the comments around quality such as "the higher variance does not lead to an improved correlation". Unless you can show otherwise I don't quite believe that you can judge absolute quality with your experimental design.

---

## Referee Comment (RC2) · E. Ray (Referee) · 1 Aug 2018

This study describes in some detail the impact of various developments of the LMDz GCM on the simulation of several trace gases. Although the new physics parameterization scheme makes the most difference in the simulated trace gases between the model cases run here, in the comparisons with observed mixing ratios there was no clear improvement in the newest versions of the model. If anything, the new physics seems to be worse at tracer transport due to the relatively weak vertical transport. This is somewhat of a null result but is important nevertheless because it shows the persistence of transport related deficiencies in model trace gas simulations. These deficiencies clearly will not be easily solved by going to higher horizontal or vertical resolution as shown here. Thus, the problem will not be fixed by more computing power.

The results are well described and organized and the topic is relevant for the journal so I recommend publication with consideration of the comments below.

Specific comments:

- Pg. 1, line 16: change "further" to "due"
- Pg. 2, line 3: remove "Already"
- Pg. 4, line 13: remove "s" from "precipitations"
- Pg. 4, line 22: "The motivation behind this change was to depart..."
- Pg. 5, line 6: looks like "1989" should be "1998"
- Pg. 8, Figure 2: The color bar is all white in my version
- Pg. 10, line 6: add "s" to end of "day"
- Pg. 10, line 12: remove "Recalling Table 1,"
- Pg. 10, line 16: "...manifests from the thermal activity that transports tracers..."

Pg. 10, line 26: the SF6 lifetime is actually most likely less than 1000 years based on a recent paper of mine (Ray et al., JGR, 2017). Not necessarily relevant to your discussion but just thought I would point it out.

Pg. 10, line 34: add "s" to end of "dynamic". Also, this result of higher stratospheric SF6 mixing ratios is quite interesting since the Brewer Dobson circulation is driven by wave activity propagating up from the troposphere. A faster circulation would require more wave activity. Yet, the decrease in deep convection seen in the Rn222 suggest less convectively driven waves so it must be from a different source, likely planetary scale Rossby waves.
Pg. 12, lines 4 and 5: remove "s" from the end of "exchanges"

Pg. 12, line 6: "...correlated with the efficiency..."

Pg. 12, line 19: remove "Recalling Table 1,"

Pg. 12, line 29: change "outreach" to "exceed"

Pg. 12, line 32: remove "while zooming over this region"

Pg. 13, Figure 4: Very difficult to see the features in the maps with such light colors.

Pg. 14, line 16: change "obtain" to "obtained"

Pg. 15, Table 2 and Section 4.2.1: Are there any measurements being compared to here? This section is a subsection under Section 4 Comparison with observations but it's not clear if the gradients shown are relative to observed MLO values or not.

Pg. 15, line 20: one of "modelled" or "simulated" should be "observed" I assume

Pg. 15, lines 20-21: what's described here is not what's shown on Figure 6. The correlations are on the top row and the ratios on the bottom.

Pg. 16, Figure 6: there's really no point in labeling the individual stations with numbers on the plots since they are too small to see. Just labeling the different colors should be sufficient. Also, there appears to be an axis label problem on 6d.

Pg. 16, line 1: "bottom" should be "top"

Pg. 16, line 4: the station names should be capitalized in the text, here and in all other places. It's much easier to identify that you are referring to a station when the name is capitalized and you have a mixture of upper and lowercase references that makes it confusing.

Pg. 17, line 2: change "there" to "they"

Pg. 17, line 4: change "over" to "out of"
Pg. 17, line 13: change "et" to "at"

Pg. 17, Figure 7: it would be helpful to make the zero line on the y axis clearer. Also, the y axis label on the bottom row should be delta CO2.

Pg. 18, line 10: if you're going to use numbers on the plot it would help to reference them by number here rather than the station abbreviation since that means the reader has to refer back to the previous figure to what numbers the abbreviations correspond to. Also, change "term" to "terms".

Pg. 18, Figure 8: it's hard to tell if any of the model runs are better than the others in this figure. Should I be able to tell?

Pg. 19, line 14: the phrase "for a little majority" is confusing

Pg. 19, line 15: change "supplementary" to "supplement"

Pg. 20, line 1: add "s" to end of "mean"

Pg. 21, line 4: remove "In parallel,"

Pg. 21, line 9: change "misfits" to another term, perhaps "bias". The term "misfits" is used in other places as well and I would recommend they all be changed.

Pg. 21, line 15: remove "an" before "enhanced"

Pg. 21, line 23: change "than" to "as"

Pg. 22, Figure 10: include the x-axis tick marks on the top of each plot so it's easier to see how the top of each profile differs from zero. Same for Figures 11-13.

Pg. 23, line 4: change to "constraints"

Pg. 24, lines 5-6: "...has not enabled the variability of the CO2 fluxes to be improved so far,..."

Pg. 24, line 20: add "in the new physics" after "5 km"
Pg. 26: I would change the Section 5 title to "Conclusions"

Pg. 26, line 3: "...benefits from a more elaborate radiative..."

Pg. 26, line 4: change "on" to "in"

Pg. 26, line 25: "...synoptic and diurnal scales..."

Pg. 27, line 8: "...latitudes further due to less..."

Pg. 27, line 12: "sensible" should be changed to "sensitive"

Pg. 27, line 13: "...enabled assessment of the quality..."

Pg. 27, line 33: change "heaviness" to "expense"

Pg. 28, line 13: remove "much"

---

## Author Comment (AC1) · 14 Sep 2018

Marine Remaud

**Correspondence:** marine.remaud@gmail.com

We would like to thank the reviewers for their careful and constructive evaluation of our paper. Please find our point-by-point reply below. The original text from the reviewers is reproduced in bold type. You will find a track of the changes attached to this document.

**1 Reply to the executive editor A. Kerkweg**

5 **As your article compares different LMDz versions please add the model name to the title in your revised submission to GMD, e.g., "On the impact of recent developments of an atmospheric general circulation model on the simulation of CO2 transport: a case study with LMDz"** We have replaced the title "On the impact of recent developments of the LMDz atmospheric general circulation model on the simulation of $CO_2$ transport" by "On the impact of recent developments of the LMDz atmospheric general circulation model on the simulation of $CO_2$ transport".

10 ## 2 Reply to Chris Jones

**As ESMs are more frequently used for climate projection and many of them include explicit simulation of CO2 transport and its 3D structure in the atmosphere, it is clearly becoming more important to adequately evaluate whether the dynamics and transport within these models is fit for purpose, and to understand the implications of any model errors or biases. C1 GMDD Interactive comment Printer-friendly version Discussion paper Over recent years, there**
15 **has not been widespread interaction between the carbon cycle/GCM/ESM community (such as C4MIP) and the trace-gas transport community such as Transcom. This paper is therefore timely and attempts to address this lack with a detailed examination of the performance of tracer transport in the LMDz GCM. The model has been developed for use in CMIP6 and although not explicitly mentioned, presumably aims to perform the C4MIP CO2-emissions-driven simulations. Its behaviour is compared against an older version, and the impacts of several developments are assessed**
20 **in some detail. I am not an expert in the details of transport schemes or their evaluation, but review this in the context of its potential usefulness for CMIP6/C4MIP climate simulations. I found the manuscript well written and clearly structured. The experiments performed are logical, well described and analysed logically and clearly. Overall I find the manuscript well suited to publication in GMD and recommend mainly minor amendments.**
**I have one specific query though. I am concerned by the apparent circularity of deriving surface CO2 fluxes using**
25 **LMDx in an inversion system and then using these fluxes as input in a forward transport experiment to test against**

**observations. Surely this is more than the passing mention in section 2.3, "... the surface fluxes carry some imprint of a**
**... [LMDz]...". You mention that other inversion products exist and could have been used - so why did you choose one so**
**closely related to the model you were evaluating? At the very least I would like to see some defence of this choice - can**
**you show and quantify the different sources available? How different are they? Some of your intro text describes how**
5   **surface CO2 fluxes are still very ill constrained in many places, so I imagine the different sources available to you may**
**well be quite different.**

Following your query, we have performed two additional simulations driven by the CarbonTracker $CO_2$ fluxes, another mainstream product that does not use the LMDz transport model (https://www.esrl.noaa.gov/gmd/ccgg/carbontracker/). We have summarized the differences between the two $CO_2$ flux datasets, CarbonTracker and CAMS, at the seasonal and annual
10   scale in Figure 1 of the supplementary material. As expected, the two $CO_2$ flux datasets are very different over poorly-observed regions, such as the boreal latitudes in summer and tropical regions over the whole year. For instance, the difference can be as large as 1 $kgC/m^2/year$ over Central Africa and Siberia in boreal summer. Noticeable differences are also identifiable in boreal summer over better-constrained areas, especially in Eastern Europe, Northern America and Eastern Asia. Although having been optimized with the same observations (to a large extent), the fluxes still differ in large parts of the globe. The underlying
15   reasons for these discrepancies are related to differences in data assimilation approaches, prior fluxes, uncertainty modelling and transport modelling. In this context, our paper contributes to quantify transport modelling uncertainties associated with specific model setups.

**I could envisage some value in using multiple (or at least 2) sources and showing an additional panel to figure 3 - how**
**does the choice of surface fluxes compare with changes in resolution and surface physics?**
20   In the supplementary material, we have reproduced most of the article figures, with the addition of the results obtained with the CarbonTracker fluxes. Figures 2 through 7 show that the impact of the change in surface fluxes is of a similar magnitude to, if not larger than, the change in the physics, and depends on the diagnostics considered.

**A few minor points are listed below, but if you can address this main concern then I would consider this a very sound**
**paper.**

25   1. **section 2.2. finishes by mentioning a mass correction - can you elaborate slightly? Is this simply a diagnoses of the**
    **loss on each timestep which is then added back evenly distributed through space?** The reviewer guessed well and
    we have added this information using the reviewer's words.

  2.  **in a couple of places you mention CMIP4. I am assuming this is meant to say CMIP5? (but could also mean**
    **CMIP3 which accompanied AR4, or it could mean C4MIP - the confusion of MIPs is endless!). Please check**
30     **and clarify.** Indeed, this is meant to be CMIP5 : CMIP4 has been replaced by CMIP5 in the article. In fact, the standard
    version of the physics called "5A" in our paper has been developed in the frame of the CMIP3 project which accompanied
    AR4 and, except tuning and an improved resolution, remained the same throughout the CMIP5 project. The new physics
    called "6A" has been meant to be used in the CMIP6 project. The reference climate configurations are summarized in
    the Table below.

|  | Physical parameterization | Horizontal grid | Number of vertical layers |
|---|---|---|---|
| CMIP3 | 5A (old physics) | $96 \times 72$ | 19 |
| CMIP5 | 5A (old physics) | $96 \times 95$ | 39 |
| CMIP6 | 6A (new physics) | $144 \times 142$ | 79 |

**Table 1.** Summary of the reference climate configurations.

3. **section 4.3 uses independent measurements of vertical profiles - this seems like a more reasonable (i.e. not circular on the choice of transport model used for the inversion that produced the surface fluxes) test. If this is really the case then I'd expect the results here to be (almost) independent of your choice of surface fluxes, so that would be a good check if you performed a run with different sources - do the results in this section still come out the same?** In the supplementary material, we have reproduced the figures from the section 4.3 and added simulations with two physical parameterizations using the CarbonTracker fluxes. The main findings remain unchanged. We have seen that the convective inhibition induced by the new physics tend to trap the anomalies within the boundary layer, resulting in a higher seasonal cycle at the surface, as emphasized in Figures 2 and 4. The vertical profiles of the $CO_2$ mixing ratio are only slightly modified by the change in the fluxes. The effect of the physics on the mean vertical profiles of $CO_2$ remains small and of the same magnitude (less than 1 ppm) over Northern America, Europe, Eastern Asia, Northern Southern Asia and remain large over Brazil (4 ppm) and over Northern America in summer (1 ppm). We mention the supplementary material in section 2.3: *In the supplement, we show the robustness of our conclusions with respect to a change of the $CO_2$ surface fluxes from CAMS to CarbonTracker (https://www.esrl.noaa.gov/gmd/ccgg/carbontracker/.*

4. **related to my main point, I'm quite happy with conclusions about the impact of processes such as "the new physics tends to weaken the vertical mixing" - this seems a reasonable conclusion from your analysis. Where I am less comfortable are the comments around quality such as "the higher variance does not lead to an improved correlation". Unless you can show otherwise I don't quite believe that you can judge absolute quality with your experimental design.** We have added the following sentence in the main text to make this clearer: *The change of physics increases the amplitude of the synoptic variability without affecting the phasing.*

**3   Reply to Eric Ray**

**This study describes in some detail the impact of various developments of the LMDz GCM on the simulation of several trace gases. Although the new physics parameterization scheme makes the most difference in the simulated trace gases between the model cases run here, in the comparisons with observed mixing ratios there was no clear improvement in the newest versions of the model. If anything, the new physics seems to be worse at tracer transport due to the relatively weak vertical transport. This is somewhat of a null result but is important nevertheless because it shows**

the persistence of transport related deficiencies in model trace gas simulations. These deficienC1 GMDD Interactive comment Printer-friendly version Discussion paper cies clearly will not be easily solved by going to higher horizontal or vertical resolution as shown here. Thus, the problem will not be fixed by more computing power. The results are well described and organized and the topic is relevant for the journal so I recommend publication with consideration of the comments below.

1. **Pg. 1, line 16: change "further" to "due"** : done

2. **Pg. 2, line 3: remove "Already"** : done

3. **Pg. 4, line 13: remove "s" from "precipitations"** : done

4. **Pg. 4, line 22: "The motivation behind this change was to depart ... "** : done

5. **Pg. 5, line 6: looks like "1989" should be "1998"** : done

6. **Pg. 8, Figure 2: The color bar is all white in my version** : done

7. **Pg. 10, line 6: add "s" to end of "day"** : done

8. **Pg. 10, line 12: remove "Recalling Table 1,"** : done

9. **Pg. 10, line 16: "... manifests from the thermal activity that transports tracers ..."** : done

10. **Pg. 10, line 26: the SF6 lifetime is actually most likely less than 1000 years based on a recent paper of mine (Ray et al., JGR, 2017). Not necessarily relevant to your discussion but just thought I would point it out.** : We have changed "over" to "around" and referenced your paper.

11. **Pg. 10, line 34: add "s" to end of "dynamic".** : done.

12. **Also, this result of higher stratospheric SF6 mixing ratios is quite interesting since the Brewer Dobson circulation is driven by wave activity propagating up from the troposphere. A faster circulation would require more wave activity. Yet, the decrease in deep convection seen in the Rn222 suggest less convectivly driven waves so it must be from a different source, likely planetary scale Rossby waves.** : we agree with this interesting point. We believe that it deserves a dedicated investigation and we have therefore not changed the text.

13. **Pg. 12, lines 4 and 5: remove "s" from the end of "exchanges"** : done

14. **Pg. 12, line 6: " ... correlated with the efficiency ..."** : done

15. **Pg. 12, line 19: remove "Recalling Table 1,"** : done

16. **Pg. 12, line 29: change "outreach" to "exceed"** : done

17. **Pg. 12, line 32: remove "while zooming over this region"** : done

18. **Pg. 13, Figure 4: Very difficult to see the features in the maps with such light colors.** : The colorbar aims at highlighting the regions that exceed values 0.3 and 0.5 ppm chosen by GHG-CCI (2016) as reference for the impact of systematic errors in the assimilation of satellite retrievals in atmospheric inversions. The light colors in most regions mean the differences are weak and that the transport (in our case) is not a limitation for inversion purpose.

19. **Pg. 14, line 16: change "obtain" to "obtained"** : done

20. **Pg. 15, Table 2 and Section 4.2.1: Are there any measurements being compared to here? This section is a subsection under Section 4 Comparison with observations but it's not clear if the gradients shown are relative to observed MLO values or not.** : Since the observed values of the gradient to MLO have been assimilated by our inversion system, the comparison with observations is not relevant here. The best agreement with observations will be systematically obtained with the standard version which has been used to optimize the surface fluxes. The purpose of the plot is to quantify the sensibility of the inversion from surface observations to model setups.

21. **Pg. 15, line 20: one of "modelled" or "simulated" should be "observed" I assume** : done

22. **Pg. 15, lines 20-21: what's described here is not what's shown on Figure 6. The correlations are on the top row and the ratios on the bottom.** : done

23. **Pg. 16, Figure 6: there's really no point in labeling the individual stations with numbers on the plots since they are too small to see. Just labeling the different colors should be sufficient. Also, there appears to be an axis label problem on 6d.** : We have kept the numbers because they enable identifying outlier sites.

24. **Pg. 16, line 1: "bottom" should be "top"** : done

25. **Pg. 16, line 4: the station names should be capitalized in the text, here and in all other places. It's much easier to identify that you are referring to a station when the name is capitalized and you have a mixture of upper and lowercase references that makes it confusing.** : We mixed upper and lower case references in order to easily identify which stations have been assimilated (as indicated in the legend of Figure 6).

26. **Pg. 17, line 2: change "there" to "they"** : done

27. **Pg. 17, line 4: change "over" to "out of"** : done

28. **Pg. 17, line 13: change "et" to "at"** : done

29. **Pg. 17, Figure 7: it would be helpful to make the zero line on the y axis clearer. Also, the y axis label on the bottom row should be delta CO2.** : done

30. **Pg. 18, line 10: if you're going to use numbers on the plot it would help to reference them by number here rather than the station abbreviation since that means the reader has to refer back to the previous figure to what numbers the abbreviations correspond** : done

31. **to. Also, change "term" to "terms".** : done

32. **Pg. 18, Figure 8: it's hard to tell if any of the model runs are better than the others in this figure. Should I be able to tell?** : The primary result that can be obtained from this figure is that the new physics increases the amplitude of the $CO_2$ synoptic variability. The reference simulation underestimates the variability. It is not possible to say which physics is better though. : some stations have gotten closer to the observations while some others clearly depart more from the observations because of an overestimated variability.

33. **Pg. 19, line 14: the phrase "for a little majority" is confusing** : We changed it to "at continental sites AMY, MVY and NGL" instead.

34. **Pg. 20, line 1: add "s" to end of "mean"** : done

35. **Pg. 21, line 4: remove "In parallel,"** : done

36. **Pg. 21, line 9: change "misfits" to another term, perhaps "bias". The term "misfits" is used in other places as well and I would recommend they all be changed.** : "bias" is a statistical value, while by "misfit" we mean individual model-data differences. We have changed "misfit" to "difference".

37. **Pg. 21, line 15: remove "an" before "enhanced"** : done

38. **Pg. 21, line 23: change "than" to "as"** : done

39. **Pg. 22, Figure 10: include the x-axis tick marks on the top of each plot so it's easier to see how the top of each profile differs from zero. Same for Figures 11-13.** : done

40. **Pg. 23, line 4: change to "constraints"** : done

41. **Pg. 24, lines 5-6: "...has not enabled the variability of the CO2 fluxes to be improved so far, ..."** : done

42. **Pg. 24, line 20: add "in the new physics" after "5 km"** : done

43. **Pg. 26: I would change the Section 5 title to "Conclusions"** : done

44. **Pg. 26, line 3: "...benefits from a more elaborate radiative..."** : done

45. **Pg. 26, line 4: change "on" to "in"** : done

46. **Pg. 26, line 25: "...synoptic and diurnal scales..."** : done

47. **Pg. 27, line 8: "...latitudes further due to less..."** : done

48. **Pg. 27, line 12: "sensible" should be changed to "sensitive"** : done

49. **Pg. 27, line 13: " ...enabled assessment of the quality..."** : done

50. **Pg. 27, line 33: change "heaviness" to "expense"** : done

5    51. **Pg. 28, line 13: remove "much"** : done

**References**

[revised manuscript text omitted]